

# Signatures of the Madden-Julian Oscillation in Middle Atmosphere zonal mean Temperature: Triggering the Interhemispheric Coupling pattern

Christoph G. Hoffmann[1], Lena G. Buth[1,2], and Christian von Savigny[1]

[1]Institute of Physics, University of Greifswald, Felix-Hausdorff-Str. 6, 17489 Greifswald, Germany
[2]now at Alfred Wegener Institute, Helmholtz Centre for Polar and Marine Research, Bremerhaven, Germany

**Correspondence:** Christoph G. Hoffmann (christoph.hoffmann@uni-greifswald.de)

**Abstract.** The Madden–Julian oscillation (MJO) is the dominant mode of intraseasonal variability in the troposphere. The influence of the MJO on the middle atmosphere (MA) and particularly on its temperature is of interest for both the understanding of MJO-induced teleconnections and research on the variability of the middle atmosphere. However, only few studies dealing with the influence of the MJO on MA temperature are available.

We analyze statistically the connection of the MJO and the MA zonal mean temperature based on observations by the MLS satellite instrument. We consider all eight MJO phases, different seasons and the state of the quasi-biennial oscillation (QBO). We show that the MA temperature is influenced by the state of MJO in large areas of the MA and under roughly all considered atmospheric conditions. The zonal mean temperature response is characterized by a particular spatial pattern, which we link to the interhemispheric coupling (IHC) mechanism, a known dynamical feature of the MA. The strongest temperature

deviations are on the order of $\pm 10\,\mathrm{K}$ and are found in the polar winter MA during boreal winter when the QBO is in the easterly phase. Other atmospheric conditions also show temperature responses with the characteristic spatial pattern, but weaker and more noisy. The QBO turns out to have a relatively big influence during boreal winter but only a small influence during austral winter. We also discuss the role of sudden stratospheric warmings (SSWs), which have an ambivalent influence on our interpretation, because they introduce strong temperature variability in the polar winter MA themselves. In addition, SSWs are

one possibility to explain the QBO influence during boreal winter. Furthermore, we also analyze the change of the temperature response pattern while the MJO progresses from one phase to the next. We find a largely systematic reaction of the MA to the phase changes, particularly a gradual altitude shift of the MA temperature response pattern, which can be seen more or less clearly depending on the atmospheric conditions.

   Overall, a major outcome of the present study is the finding that the tropospheric MJO can trigger the IHC mechanism, which

affects many areas of the MA. It is therefore a noteworthy example for the complex couplings across different atmospheric layers and geographical regions in the atmosphere. Additionally, it highlights close linkages of known dynamical features of the atmosphere, particularly the MJO, the IHC, the QBO, and SSWs.

   Because of the wide coverage of atmospheric regions and included dynamical features, the results might help to further constrain the underlying dynamical mechanisms and could be used as a benchmark for the representation of atmospheric

couplings on the intraseasonal timescale in atmospheric models.



# 1 Introduction

The Madden-Julian-Oscillation (MJO), first documented by Madden and Julian (1972), is known as the dominant mode of intraseasonal variability in the tropical troposphere (Zhang, 2005). The major directly observable feature is a center of anoma-
lously strong deep convection, flanked by regions of reduced convection. This structure appears periodically over the Indian Ocean and travels eastward over the maritime continent until it disappears over the Pacific Ocean. The MJO is a highly variable feature of the atmosphere and, e.g., its period varies strongly between about 30 and 90 days (Zhang, 2005). Its appearance influences weather patterns in many equatorial regions, e.g., the monsoons in Asia and Australia (Zhang, 2005).

Although the obvious convective disturbance is limited to the Indo-Pacific region, dynamical anomalies, e.g., in terms of
vorticity, are measured around the whole equator (e.g., Cassou, 2008). Moreover, the MJO has also influences around the whole globe and is itself influenced from extratropical regions making the MJO a component of global teleconnection patterns (Lau and Waliser, 2012, Chapter 14). A prominent example in the northern hemisphere is a link between the MJO and the North Atlantic Oscillation (Cassou, 2008). Affected are different parameters of the polar climate system like the Arctic surface temperature (Yoo et al., 2012) or even the Arctic sea ice concentration (Henderson et al., 2014). For a review of the observed
teleconnections between the tropics and the polar regions on different time scales see, e.g., Yuan et al. (2018).

Due to the intraseasonal time scale on which the MJO acts, a deeper knowledge on the processes that control it is expected to improve the forecast skills of longer-term weather forecasts (e.g., Zhang, 2013). The understanding of the mechanisms of the teleconnections is important to improve the representation of the MJO in models but might in turn help to improve the forecasts skills also in the extratropics, e.g., in terms of the prediction of extreme weather events (Lau and Waliser, 2012, Chapter
14). Mechanisms for the teleconnections with the extratropics are reviewed by, e.g., Yuan et al. (2018). Inner-tropospheric interactions are based on the excitation of Rossby waves by the strong convection, which is associated with the MJO and propagation of zonal mean zonal wind anomalies, which are triggered by the MJO.

A further mechanism for MJO-related teleconnections is proposed by Garfinkel et al. (2012) for the extended boreal winter season. It also considers the propagation of Rossby waves, but this time not only the horizontal poleward propagation but also
the vertical propagation into the stratosphere. These waves could influence the polar vortex and modulate the appearance of midwinter major sudden stratospheric warmings (SSW), so that SSWs follow certain MJO phases. It was already known that the occurrence of sudden stratospheric warmings (SSW) during Arctic winter can then influence the tropospheric state below (e.g. Baldwin and Dunkerton, 2001). The complete chain can therefore lead to an influence of the MJO on the polar troposphere mediated by the stratosphere with a longer time scale (one to two months) than the inner-tropospheric connections.

There is also increasing evidence that the variability of the MJO itself is partly controlled by the stratosphere. A prominent example for this is the influence of the quasi-biennial oscillation (QBO; see Baldwin et al. (2001) for a review of the QBO) during boreal winter: Yoo and Son (2016) found that the MJO amplitude tends to be larger during the QBO easterly phase and smaller during the QBO westerly phase and several studies followed up on this (e.g., Son et al., 2017; Marshall et al., 2017;





Zhang and Zhang, 2018; Densmore et al., 2019; Wang and Wang, 2021). The influence on the MJO from above is also covered
by a more general recent review on the stratospheric influence on the tropical troposphere by Haynes et al. (2021).

Consequently, there is increasing awareness for the potential of considering troposphere-stratosphere couplings in both
directions in weather prediction systems, as shown by, e.g., Domeisen et al. (2020). And at least from these examples, it
becomes clear that it is of importance to consider at least the stratosphere to completely understand the functioning and
interactions of the MJO in the climate system.

In addition to this troposphere-related motivation, there is a second major motivation to study the connection of MJO and
the middle atmosphere (MA), which arises from the interest in the variability of the MA itself. One important aspect of MA
research is the determination of a long-term trend in MA temperature, which is an important indicator of anthropogenic climate
change (e.g., Randel et al., 2009; Santer et al., 2013; Maycock et al., 2018; Beig et al., 2003; Beig, 2011). Other aspects are,
e.g., the understanding of the implications of solar variability on the climate system (e.g., Gray et al., 2010), the influence of
large volcanic eruptions (e.g., Timmreck, 2012), and of course the monitoring of stratospheric ozone. All these aspects have
in common that they are connected to temporal variability of different MA parameters, with temperature being an essential
one. A major difficulty is the disentanglement of all the sources of temperature variability and the unambiguous attribution
to the respective causes to understand the individual processes separately. In order to increase the robustness of analyses and
interpretations, also the smaller sources of temperature variability should be known and quantified.

The MJO has, to our knowledge, gotten only little attention as a one possible independent source of temperature variability
in the MA. In addition to the already mentioned analysis by Garfinkel et al. (2012), the studies by Yang et al. (2017, 2019) are
in this context of relevance for our analysis. These publications cover the MJO influence on the MA temperature among other
parameters for different geographical regions, seasons and atmospheric conditions. They are mostly based on modeled and
reanalyzed data, whereas it is our aim to provide a purely observational perspective. Sun et al. (2021) analyze the effect of the
MJO on the northern mesosphere during boreal winter. They also mostly rely on modeled data but use satellite observations as
support, particularly, the same observational dataset as we use in the following (Sect. 2.1). There are more studies relevant in
the broader context, but either they treat the influence on temperature not as the main point (e.g., Moss et al. (2016); Tsuchiya
et al. (2016); Wang et al. (2018a), which focus on the MJO influence on wave activity) or are limited to different atmospheric
regions (e.g., Yang et al. (2018); Kumari et al. (2020, 2021), which discuss the MJO influence on tides in the mesosphere /
lower thermosphere region). The relationships to the relevant studies are discussed in more detail below (Sect. 4).

We note as an additional aspect for the motivation that the occurrence rates of individual MJO phases appear to be subject
to climate change (Yoo et al., 2011). Therefore, a linkage of the MJO and the MA could represent an additional pathway of
anthropogenic climate change into the MA. Conversely, the characterization of the MJO-MA relationship might also be helpful
to understand climate change related changes in the teleconnection patterns mentioned above.

All the above-mentioned aspects motivate the exploration of already existing observations with respect to the influence of
the MJO on the MA, particularly on the MA key parameter temperature. However, the mentioned publications on the MJO
influences on the MA temperature mostly deal with modeled or reanalyzed data, which has the advantage that respective mech-
anisms can be more easily identified in these more comprehensive datasets. Contrarily, this publication aims at contributing a





global quantification of the influence of the MJO on MA temperatures based observed data, to gain additional independent and

purely observationally determined information as complement to modeling efforts.

Most related studies first determine the MJO phase with the strongest response and concentrate in the following analysis mostly on this phase (e.g., Garfinkel et al., 2012; Yang et al., 2017). The temporal evolution of the atmosphere is then considered in time lags around the appearance of this phase. We consider instead all eight MJO phases, so that the transition of the MA temperature response from one MJO phase to the next becomes visible. Furthermore, we concentrate in this paper on

the analysis of zonal mean temperatures for the sake of brevity. Nevertheless, the dataset provides also a wealth of three-dimensionally resolved information, which we intend to present in a later paper.

The present paper contains a description of the datasets and methodology in Sect. 2. The results, i.e. the zonal mean MA temperature responses to the individual MJO phases for different atmospheric conditions, are shown in Sect. 3. Section 4 contains an extensive discussion, in which the results are related to known dynamical features of the MA, before we conclude

the paper in Sect. 5

## 2    Datasets and analysis approach

### 2.1    Datasets

We analyze the temperature data product measured by NASA's Microwave Limb Sounder (MLS) on the Aura satellite (Schwartz et al., 2020), version 5. A previous version of the dataset has been validated by Schwartz et al. (2008). Differences to the new

version are mentioned in Livesey et al. (2020). Data screening and exclusion has been performed according to all suggestions in the MLS quality document (Livesey et al., 2020), including the screening for cloud effects based on the MLS ice water content (IWC) product. We analyze the complete reasonable vertical range from 261 hPa to 0.00046 hPa and also use the complete temporal coverage of the ongoing measurements of about 17 years at the time of the analysis (approximately August 2004 to September 2021). The data has a spatial resolution, which roughly decreases with height in a range from about 4 km vertical

and 170 km horizontal resolution at 261 hPa to 13 km vertical and 316 km horizontal resolution at 0.00046 hPa (Livesey et al., 2020). The data has been analyzed on the original pressure grid. One has to keep in mind that the analysis grid is finer than the varying vertical resolution and that adjacent altitudes are not completely independent of each other. The original data follows the satellite track and has been averaged to fit onto temporally and horizontally regular grids. Particularly, for this paper we apply a zonal averaging with 10° resolution in the latitudinal direction.

For the characterization of the MJO, we use the OLR-based MJO Index (OMI; OLR stands for outgoing longwave radiation) introduced by Kiladis et al. (2014). We calculate the OMI values from OLR using the open source OMI calculation package (Hoffmann et al., 2021; Hoffmann, 2021), version 1.2.2. According to Hoffmann et al. (2021) a high agreement between with the original calculation routine, which is close to being identical, is expected. For the calculation, interpolated OLR data according to Liebmann and Smith (1996) has been used. From the two OMI index values per time step, the phase and the

strength of the MJO for each day can easily be calculated following the phase diagrams in Kiladis et al. (2014) and Wheeler and Hendon (2004).



We note that the MJO is subject to a seasonal variability (e.g., Zhang, 2005). The common MJO concept with the characteristic eastward propagation applies best for boreal winter. During boreal summer, the propagation direction includes a northward component (e.g., Wang et al., 2018b) and the feature is often given a distinct name, the boreal summer intraseasonal oscil-
lation (BSISO). This has also implications for the definition of appropriate MJO indices and partly particular BSISO indices are used for analyses with a special focus on boreal summer (e.g., Kikuchi et al., 2012). However, since our analysis is not restricted to boreal summer and Wang et al. (2018b) states that the MJO index OMI is also capable of reasonably tracking the BSISO, we apply OMI consistently for all seasons. This has the advantage of comparability of the results for different seasons. Nevertheless, we might check our boreal summer results with a special BSISO index in future.

For the characterization of the QBO, we use monthly mean zonal wind data above Singapore at $50\,\mathrm{hPa}$ (Naujokat, 1986). We apply the most simple discrimination approach: positive zonal wind values indicate a QBO westerly phase and negative values a QBO easterly phase.

## 2.2 Approach

Our analysis approach is a superposed epoch analysis (SEA) also known as composite analysis. The complete analysis chain
consists, however, of several steps, which are computed for each geolocation of the previously mentioned regular spatial grid separately.

First, a temperature anomaly is computed by applying a boxcar running average with a window length of $90\,\mathrm{d}$ as a low-pass filter and calculating the difference between the original and the low-pass filtered time series. Afterwards the anomaly is smoothed with a 10-day running average. The resulting anomaly time series contains therefore only strongly damped variations
on much longer (e.g., the seasonal cycle) and much shorter (e.g. weather fluctuations) time scales than the MJO, but still captures a broad range of frequencies in the intraseasonal time scale, relevant to the MJO.

Second, the data is selected according to the environmental conditions. We take generally only days into account, during which the MJO strength was greater than 1. This is a common choice in many MJO related studies to make sure that there is actually an MJO signature apparent on all considered days. Depending on the particular experimental setups described below,
the data is also selected with respect to particular seasons and with respect to the state of the QBO. Specifically, all days, which do not match the wanted setup criteria, are removed from the dataset prior to the SEA. We consider the seasons boreal winter / austral summer (December, January and February) and boreal summer / austral winter (June, July, August).

Third, the actual SEA is carried out: the selected temperature observations are grouped by the related MJO phases of each day and then averaged, so that one temperature mean value per MJO phase is calculated, complemented by the corresponding
standard errors of the mean. Note that we apply a small correction after this basic step: a mean over the complete data subset at the respective geolocation (i.e., independent of the MJO phase) is subtracted from all eight average values. This overall mean value is usually close to $0\,\mathrm{K}$, because the whole calculation is carried out on temperature anomalies, which are scattered around zero. However, the average of the selected subset of the data may slightly deviate from zero, which would complicate the comparison of the results for different geolocations if not corrected for. Fig. 1 shows two examples of the SEA results,
which represent a strong and a weak response. The number of days that go into the individual averages depends of course



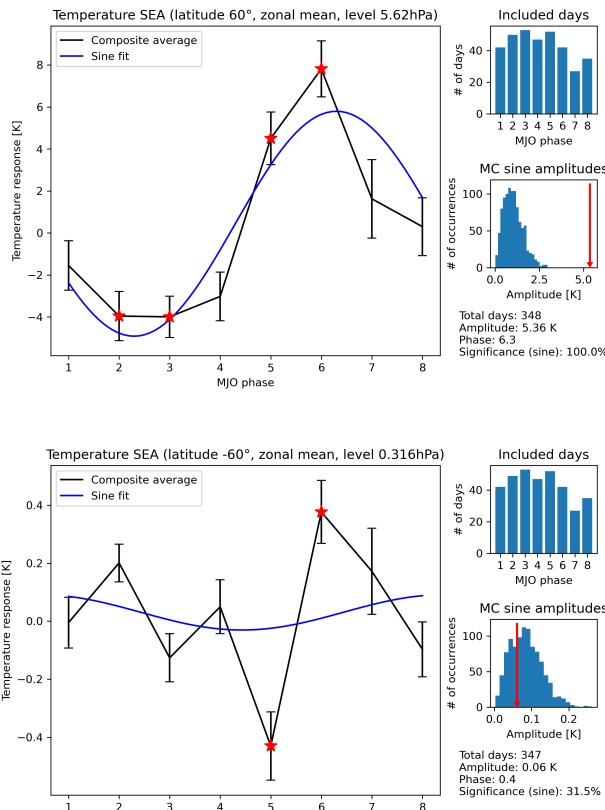

**Figure 1.** Examples of the temperature responses to the eight MJO phases at two geolocations for the boreal winter / QBO easterly situation. The examples have been selected for demonstration purposes to represent a particularly strong response (upper half) and a weak response (lower half). The following description applies to both cases: The main results, the average temperature anomalies for each MJO phase, are shown as the black lines in the larger panels together with the eight standard errors of the mean shown as error bars. The blue line shows the fitted sine curve. Temperature responses, which are significant according to the MCIP method (see Sect. 2.2 for details on the MC quantification) are marked with red stars. Results according to the MCS significance estimation method, which estimates the significance of a systematic variation over all eight phases, are shown in the lower smaller panels on the right, respectively: The blue bars constitute a histogram of all sine amplitudes derived based on random data, whereas the sine amplitude of the real data is shown as a red arrow. The resulting percentage of random amplitudes lower than the real one is also given in the lower right text field. The upper smaller panels on the right simply indicate how many days of the data went into each of the eight temperature anomaly average values.

strongly on the selection criteria, but also somewhat on the geolocation and the MJO phase itself. Rough numbers are about 400 days per MJO phase if only the MJO strength filtering is applied, 100 days with additionally the seasonal selection and 50 days with a combined seasonal and QBO selection.





As stated before (Sect. 1), we consider all eight phases of the MJO so that the transition of the MA temperature response
from one phase to the other becomes visible. At least in an ideal case, the response could vary like sine function over the course
of the eight MJO phases. Following this notion, we fit a sine function to the eight mean values as the forth analysis step to
further characterize the behavior across the MJO phases. The parameters amplitude, phase and offset can freely be adjusted by
the fit, whereas the period is fixed to 8 MJO phases. The fit results can be analyzed with different foci. However, in this study
only the resulting amplitudes are used for a significance estimation in the next analysis step. The two examples in Fig. 1 also
show the fitted sine curves. It is seen that the strong response exhibits indeed a sine-like behavior and the amplitude roughly
represents the strength of the deviations, whereas the appearance of the weak response is more noisy, which results in a even
weaker sine amplitude.

The fifth and final step is a significance estimation using a Monte Carlo (MC) method. For this, the analysis is basically
repeated multiple times, each time with randomly modified input data. The resulting distribution of artificial results can be
used to estimate how likely a particular temperature response can be the result of random fluctuations instead of physical
reasons. As we have pointed out in Hoffmann and von Savigny (2019), there is a wide scope of individual decisions in the
design of the MC calculations, which can influence the final significance estimation. In order to not distract the reader from
the basic results, we concentrate here on only one version of the random data generation (as most other publications also do),
which is well comparable to many previous publications. Particularly, we randomly redistribute the MJO index time series 1000
times, i.e. the attribution of MJO phases and strengths to the individual days of the time series is changed. The temperature
data remains thereby untouched. The SEA calculation is repeated for each random data realization resulting in a distribution
of possible temperature responses. Whereas we only present one kind of random data generation, we still show two different
kinds of the final quantification of the significance. The first one, abbreviated as MCIP for "Monte Carlo Individual Phases"
in the following, is also best comparable to previous studies, particularly if they are focused on individual MJO phases. It is
simply checked for each MJO phase separately if the absolute value of the real SEA anomaly result for a particular MJO phase
is greater than $95\,\%$ of the absolute values of the responses based on random data. Hence, this significance estimation checks
if the derived SEA temperature anomaly is strong enough to be only very unlikely produced by random fluctuations for each
MJO phase separately. Significant average values according to this method are marked with a red star in Fig. 1. An advantage
is that this significance estimation exists separately for each MJO phase. However, this method has the disadvantage that it
may underestimate significance. E.g., if a we consider the result for MJO phase 1 in the upper panel of Fig. 1, the response
is relatively close to $0\,\mathrm{K}$ and consequently also likely reproduced by random data and therefore not marked as significant.
However, the development of the response over all eight MJO phases indicates that the response of phase 1 could be a part
of an overall systematic variation, albeit approximately at the zero-crossing of the variation. Therefore, we use a second
quantification approach, abbreviated as MCS for "Monte Carlo Sine" in the following, in which it is checked that the real
amplitude of the fitted sine function is greater than $95\,\%$ of the fitted amplitudes based on the random data. Hence, this
quantification approach evaluates the systematic behavior over all 8 phases and is therefore also based on a larger number
of samples (all days that go into the eight averages instead only those, which go into one specific average). Examples of the
distributions of sine amplitudes are also given in the lower smaller plots in Fig. 1.





## 3   Results: MJO influence on zonal mean MA temperature

This study is focused on the zonal mean response of MA temperature to the MJO. Nevertheless, a quick inspection of the full three-dimensional structure of the response exhibits also zonal structures, particularly in the lower stratosphere. These structures partly cancel out by the zonal averaging so that the zonal mean anomaly values shown in the following are likely lower than some of the individual local responses.

### 3.1   No filtering for environmental conditions

As will become obvious in the following, external conditions like the season and the QBO have a strong influence on the strength and spatial structure of the response. Hence, analyzed data that comprises different states of these conditions might also result in reduced responses due to interferences of the individual responses to these conditions. Therefore, we do not discuss the unfiltered analysis here in detail. Instead, we start directly with the boreal winter situation in the following (Sect. 3.2). Nevertheless, we show the results for the unfiltered data (only a filtering for MJO strength greater than 1 is applied) in the
supplement (Fig. S1) and note here that the derived temperature anomalies connected to the MJO influence are in this case in the range of $\pm 2\,\mathrm{K}$.

### 3.2   Boreal winter / austral summer

We start with the data restricted to boreal winter, since it is known that the strongest signal is likely produced by the MJO during that season. Fig. 2 shows the meridional plane temperature responses for each of the eight MJO phases. The strongest
response is seen for the MJO phases 5 and 6 in the northern polar region, i.e., in the winter hemisphere. This response is spread vertically and divided into two zones with opposite sign visible as strong red and blue areas in the figure. The departures from the mean state are as high as $\pm 6\,\mathrm{K}$ even in this zonal mean picture and are therefore noticeably higher than for the unfiltered case mentioned above (Sect. 3.1).

A broader overall pattern in the meridional plane response can be recognized more or less for each MJO phase. However,
it is also clearest for MJO phase 5, so that this phase serves as an example and is shown again with some visual guidance in Fig. 3 to support the following description: The two strong anomalies over the winter pole described before (indicated with dash-dotted circles in Fig. 3) constitute a vertical dipole with the positive anomaly around $3\,\mathrm{hPa}$ and the negative anomaly around about $0.03\,\mathrm{hPa}$. This dipole is part of a remarkable global structure, which comprises three more zones of spatially coherent anomalies, which are described in the following. The departures from the mean state for these three zones are with
less than roughly $1\,\mathrm{K}$ much weaker than the polar ones, but are still mostly significant, particularly when considering the MCS significance estimation. Two of these weaker anomalies constitute another vertical dipole in similar altitudes as the first one, but located around the equator (indicated with dashed circles in Fig. 3). This equatorial dipole has, however, opposite sign compared to the polar dipole, so that all four zones together look like quadrupole spanning over the complete winter MA. Put in other words, areas of mutually opposing responses between the stratosphere and the mesosphere as well as the equatorial
and the polar winter latitudes can be identified. Whereas these four zones are roughly found in the winter hemisphere, the fifth





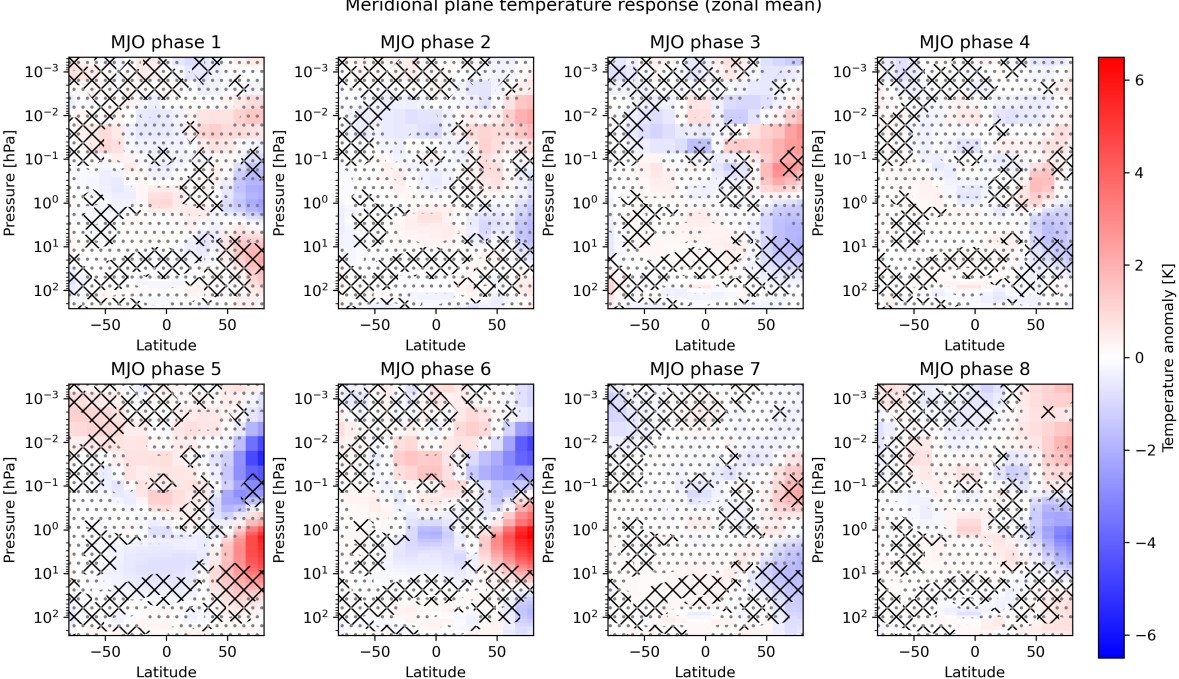

**Figure 2.** Meridional plane temperature responses for all eight MJO phases for the boreal winter / austral summer situation. Each panel contains the temperature anomaly response to a particular MJO phase for all geolocations in the zonal mean. All eight panels share the same color scale on the right. Insignificant values according to the MCIP method are marked with gray dots. Insignificant areas according to the MCS method are marked with black crossed lines. Note that the insignificance pattern according to the MCS method is identical for all 8 MJO phases, since the overall systematic behavior is evaluated (slight differences of the hatches between the different panels are due to numerical inaccuracies in the rendering of the image and do not indicate differences in the significance estimation results).

response zone is in the summer polar latitudes (indicated with a solid circle in Fig. 3). It is even higher up between $0.01\,\mathrm{hPa}$ and $0.001\,\mathrm{hPa}$ and its extent is much smaller than that of the first four zones. It shows a positive anomaly and looks like a polar summer mesosphere extension of the positive equatorial anomaly around $0.1\,\mathrm{hPa}$. As we will come back to this pattern several times, we will call it "five-zone-response" for a clearer identification in the following. Its origin will also be discussed below

in Sect. 4.1.

A comment on the appearance of the strongest response particularly in the northern winter middle atmosphere is due, since this is a region that is in any case characterized by strong variability. This strong variability is connected to the properties of the polar vortex and particularly the appearance of SSWs (e.g., Baldwin et al., 2021). Whereas the SEA method applied here is designed to average out all variability that is not correlated with the MJO phase evolution, the success of this elimination

depends on the length of the analyzed time series and the magnitude of the unrelated variability. Hence one might argue that the



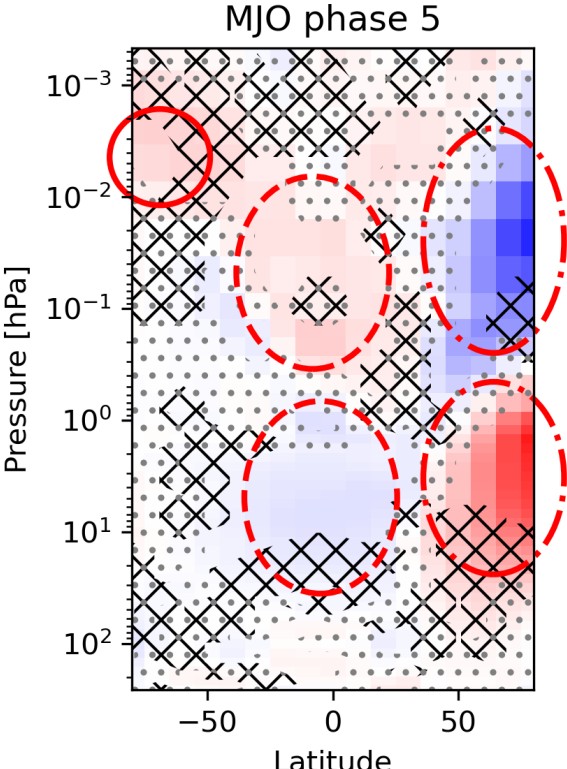

**Figure 3.** Repetition of the data shown for MJO phase 5 in Fig. 2, but with the five-zone-response highlighted to guide the reader better through the written description. The two dash-dotted circles indicate the two zones of the polar winter dipole in the temperature response. The two dashed circles mark the corresponding equatorial dipole, which has a reversed sign compared to the polar dipole. The solid circle marks the fifth anomaly zone located in the summer mesosphere, which has the same sign as the upper equatorial anomaly. See Sect. 3.2 for details.

analyzed time span of 17 years is not enough for the elimination of the strong uncorrelated polar winter variability, so that the method produces false responses of temperature to the MJO in the polar winter region. While we can indeed not totally exclude that uncorrelated variability still influences the results, we have several indications that the strong polar signal is overall truly correlated with the MJO phases. First, there are indications that the occurrence of SSWs themselves is correlated to the MJO

(Garfinkel et al., 2012). Hence, the results presented here must not necessarily be interpreted in a way that they are compromised by the strong SSW variability, but they could also be interpreted as supportive for the idea of an interaction between the MJO and SSWs. Second, as will be discussed in Sect. 3.5, a similar pattern exists also for the southern hemisphere polar winter, which is dynamically much more quiescent. Third, although much weaker, also the polar summer hemisphere temperatures show partly responses to the MJO, see e.g., MJO phase 2 in Fig. 2 between $10\,\mathrm{hPa}$ and $0.1\,\mathrm{hPa}$. Hence, there seems so be a

route for MJO variability into the polar middle atmosphere independent of vortex dynamics. As a forth argument, one could





consider repeating the analysis only for boreal winters without SSW. We have quickly checked this approach, which indicated that the signature is still clearly visible. However, the number of days that go into the analysis is quite low, when applying this additional filter so that the interpretation needs particular care and we have excluded this from the current analysis. All in all, we think that at least the baseline of the strong polar winter response is actually truly connected to the MJO. The full strength is then probably result of an amplification by an interaction of MJO and SSWs, although we cannot totally exclude that it is partly also caused by uncorrelated strong variability.

The five-zone-response appears more or less intense for each of the MJO phases while it has an opposite sign for some of the phases (compare, e.g., the responses for the phases 6 and 2 in Fig. 2). Of interest could therefore also be the transition of the response from one MJO phase to the next to get an idea of possible temporal systematics. For some of the phases, a gradual systematic change of the response is indeed recognizable, e.g., a downward shift of the polar winter dipole from MJO phase 2 to phase 4. Furthermore, some of the opposite MJO phases show also temperature responses with an opposing sign, as expected for an ideal temperature oscillation during the course of one MJO cycle (e.g., the previously mentioned phases 2 and 6). However, the mapping of the responses of opposite phases is not perfectly symmetric. E.g., the phases 5 and 6 appear to be opposed by the phases 2 and 3 instead of 1 and 2 and the responses of the opposite phases 3 and 7 show roughly no sign switch at all. We will elaborate on the aspect of the phase transition in the context of the QBO influence in Sect. 3.3.

### 3.3 Boreal winter and QBO easterly

Fig. 4 shows a similar analysis but with data selected for boreal winter and QBO easterly conditions. Overall, the responses look similar to those for boreal winter data in Fig. 2 (note that the color scale has a different range). Particularly the five-zone-response is recognizable for each MJO phase, although the responses for MJO phases 1 and 8 look somewhat disturbed (which will become part of the interpretation below). The significant part of the summer mesosphere extension appears generally to be higher up at around $0.001\,\mathrm{hPa}$ for the boreal winter and QBO easterly situation.

It is obvious that the temperature anomalies are even stronger than for boreal winter filtering alone. The strongest response is again seen in MJO phase 6, here now on the order of $\pm 10\,\mathrm{K}$. This suggests that the state of the QBO has an important influence on the propagation of the MJO signal into the MA during boreal winter.

Also the transition of the response pattern from one MJO phase to the next becomes clearer when the data is additionally filtered for QBO easterly conditions and we will discuss the main aspects in the following. First, it appears that the responses of the MJO phases 2 to 4 look quite similar. And at least for the MJO phases 2 and 3 the sign of the response of their opposing phases 6 and 7 is switched, as might be expected for an ideal conception. These phases 6 and 7 are also part of three connected phases, which have the same sign of the anomaly pattern: the phases 5 to 7. Hence, the response of the phases 2 to 4 seems to be opposed by the phases 5 to 7, which does not totally comply with the ideal case, in which phases 6 to 8 would be expected as opponents of the phases 2 to 4. It is also seen, that the polar winter dipole response seems to descent from phase 5 to phase 7, which is not clearly seen for phases 2 to 4. It is remarkable that the only gradual variations between phases 2 to 4 on the one side and 5 to 7 on the other side are divided by an abrupt switch of the sign from phase 4 to phase 5, between which the response pattern roughly completely reverses. Hence, at first sight one interpretation of the phase transition scheme could



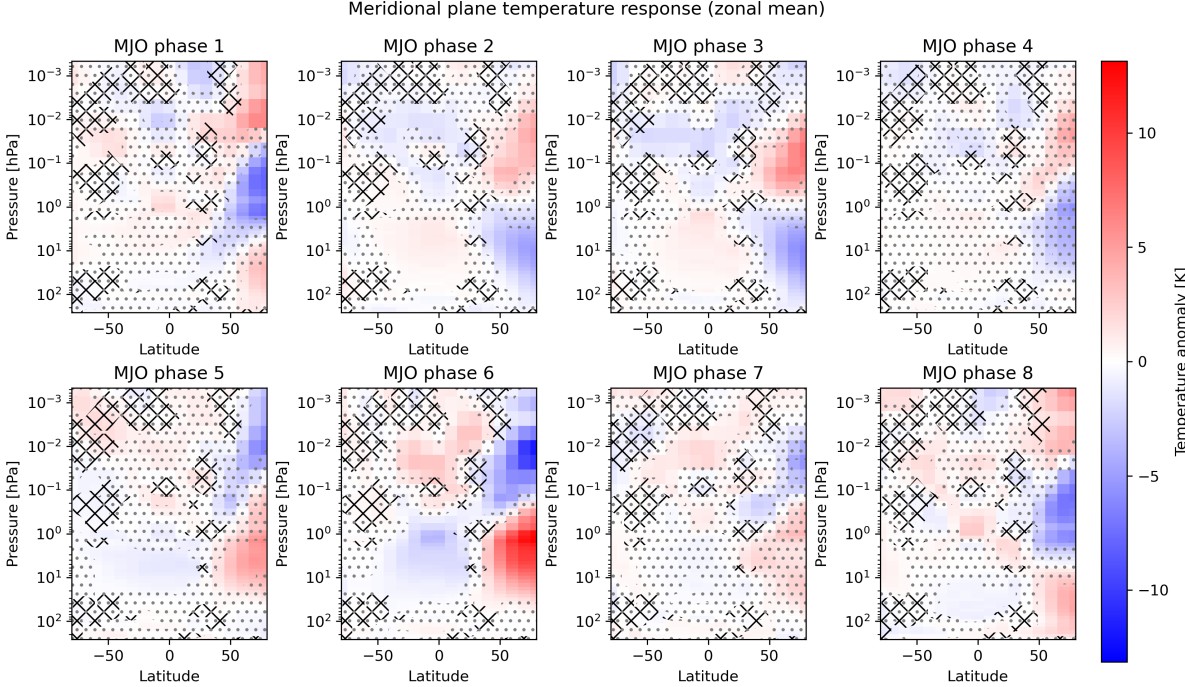

**Figure 4.** Similar to Fig. 2 but for the boreal winter / austral summer and QBO easterly situation. Note that the color scale is different compared to Fig. 2

.

be that there are two slowly varying states of MA temperature, between which an abrupt switch occurs. This suggests that there should be a switch back around the MJO phases 8 and 1, but the responses of these phases are a bit more complicated to interpret, because a third pronounced and significant anomaly zone appears above the pole and extends the polar dipole. Nevertheless, if one assumes a descent of the negative (blue) polar temperature anomaly from phase 7 to 8 and extends this in a cyclic manner to phase 1 and 2, a not-so-abrupt backward transition could be perceived here. However, if the idea of a

continuously descending pattern is extended to all phases, a second interpretation appears more probable: It is most easily seen by following the negative polar winter anomaly with the eyes and starting with MJO phase 5 (blue area in the upper right of the MJO phase 5 panel in Fig. 4): The negative anomaly is there in the highest altitudes and then descends in the following MJO phases including 8 and 1 until it reaches the lowest altitude in MJO phase 2. After phase 2 the transition comes slowly to an end, which is finally reached in MJO phases 3 and 4. Afterwards, the situation is abruptly reset during or after phase 4

and starts from the beginning with the reversed pattern in the new phase 5. In this picture, the additionally appearing second polar positive anomaly areas during the phases 8 and 1 in the highest altitudes could be seen as replacement for the vanishing




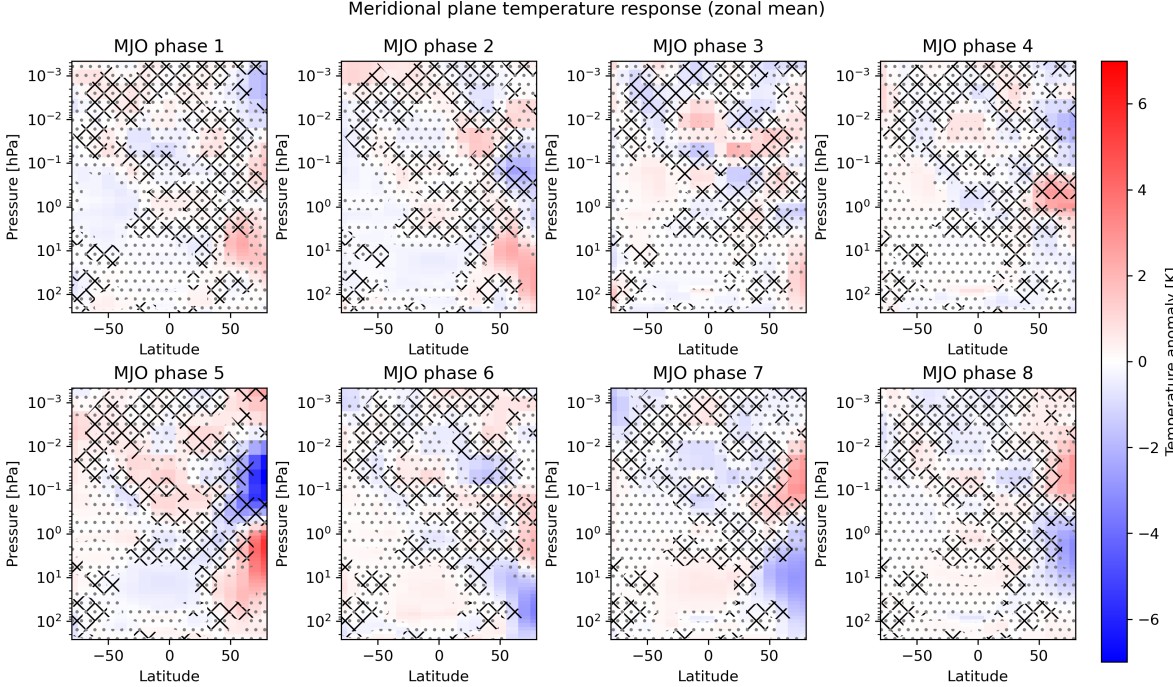

**Figure 5.** Similar to Fig. 4 but for the boreal winter / austral summer and QBO westerly situation. Note that the color scale is different compared to Fig. 4

.

positive anomaly in the lowest altitudes. Note that the summer mesosphere extension of the five-zone-response remains at its altitude during all eight MJO phases.

## 3.4 Boreal winter and QBO westerly

We have also checked the MA temperature response for the boreal winter and QBO westerly situation and find that it is much less clear than in the boreal winter and QBO easterly case (Sect. 3.3). This meets the expectations as will be further discussed in Sect. 4.4. Nevertheless, the five-zone-response is also recognizable for some of the MJO phases (Fig. 5). Also other features like the third polar anomaly zone above the polar dipole are visible again (MJO phase 5 in Fig. 5). The strongest response appears in MJO phase 5 and is on the order of $\pm 6\,\mathrm{K}$, which is the same order of magnitude as for the boreal winter situation
without QBO filtering. Therefore, the MJO influence on MA temperature has not totally vanished and has probably also to be considered under QBO westerly conditions. However, due to the more noisy spatial pattern and the somewhat smaller amount of significant locations, it appears reasonable not to discuss Fig. 5 in detail. We note that the structure might become clearer in the future, when longer observational records are available.





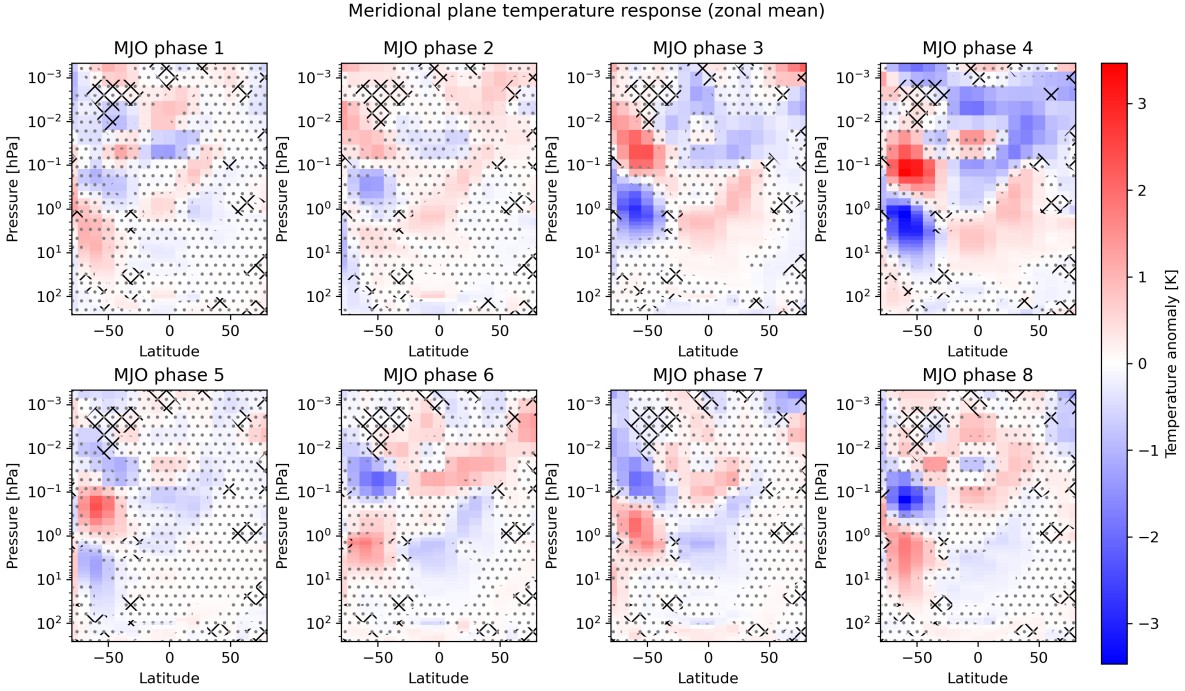

**Figure 6.** Similar to Fig. 2 but for the austral winter / boreal summer situation. Note that the color scale is different compared to Fig. 2
.

Still, we would like to mention that the responses to the individual MJO phases have roughly the opposite sign compared
to the respective MJO phase for QBO easterly filtering, although shifts in the altitudes of the response features make a 1-to-1
attribution difficult (compare, e.g., phases 2 and 7 in Fig. 4 and Fig. 5). The MJO phase 5 appears to have the same special role
as phases 1 and 8 for QBO easterly filtering. Note that the appearance of this feature is also shifted by about half of an MJO
cycle.

### 3.5 Austral winter / boreal summer

The analysis of the data restricted to austral winter (Fig. 6) shows generally that the previously described boreal winter re-
sponses can also be detected in the southern hemisphere, although they are weaker and more patchy. The strongest anomalies
are again found in the polar winter regions, but have with $\pm 3\,\mathrm{K}$ only half of the magnitude compared to the boreal winter data,
which has not been filtered for the QBO (Fig. 2). The polar winter response shows again the pattern of a vertical dipole, al-
though individual exceptions like MJO phase 5 exist, in which a third zone appears as described before. The equatorial vertical
dipole is partly somewhat more difficult to identify but is still clearly visible for many MJO phases, at least for the phases 3,
4, 6, 7, and 8. Also the summer mesosphere extension, this time in the northern hemisphere of course, can be identified in the





response of most MJO phases, e.g., for MJO phase 6. However, the summer mesosphere response is more complex for some of the MJO phases: it has a dipole structure itself (e.g., MJO phases 3 and 7). It is expected from the previous results that the summer mesospheric anomaly has the same sign as the upper part of the equatorial dipole, which is actually seen also for these phases. But in addition, we find another anomaly zone above, which has the opposite sign, forming a summer mesospheric dipole. It is remarkable that this dipole also reverses the sign for the opposing MJO phases 3 and 7, which suggest more a systematic behavior rather than a random effect.

The two significance estimation approaches provide quite different results: the MCIP method shows less and smaller significant areas, which is connected to the generally weaker response. The MCS suggests instead a significant systematic response over all eight phases in almost all places of the meridional plane. Hence, although the responses are weak in terms of the individual temperature anomalies, they appear to be clearly systematically correlated with the eight MJO phases. This might be connected to the fact that this season is dynamically less disturbed so that less interfering disturbances mask the pure MJO signal.

We note that it is not necessarily expected that the austral winter response is recognizable as a counterpart of the boreal winter response and its existence appears noteworthy on its own. This is also because the austral winter hemisphere is usually dynamically much more quiescent than the boreal winter hemisphere. In particular, (major) SSWs are very rare events in the southern hemisphere and only two such warmings have been observed since routine observations started some decades ago, namely in the austral winters of 2002 and 2019 (e.g., Jucker et al., 2021; Allen et al., 2003; von Savigny et al., 2005). While the first one is not included in the MLS period anyway, we have repeated our analysis with the MLS dataset restricted to the period before the end of 2018, so that also the second warming is not included (Fig. S2 in the supplement). It turns out that the pattern of the temperature response to the MJO phases remains basically unchanged when this warming is excluded. In contrast to the expectations, the strongest anomalies are even a bit stronger without 2019. This shows that the appearance of SSWs is not a precondition for the MJO-induced temperature response pattern. A fact that was unclear from analyzing the boreal winter response alone as discussed before (Sect. 3.2). However, due to the different magnitudes of the responses, the results still indicate that the dynamical disturbances of the boreal winter hemisphere are important for an amplification of the MJO signal.

When comparing the boreal and austral winter five-zone-response, it becomes evident that the sign of the response is identical for most individual MJO phases (e.g., for MJO phase 3 the polar negative anomaly is located below the positive anomaly in both Fig. 2 and Fig. 6 and so forth). This is particularly true, when comparing to the boreal winter and QBO easterly situation (Fig. 4), which shows a higher level of structure as discussed before. Hence, the phasing of the response with respect to the MJO trigger appears to be generally similar for both hemispheres (whereas it was unclear if it is maybe changed by the QBO westerly phase during boreal winter, as discussed in Sect. 3.4)

Also the phase transition shows similarities with the boreal winter situation and particularly the boreal winter and QBO easterly situation: Again the responses of the phases 2 to 4 are quite similar and are opposed by the responses of the phases 6 to 8, which also show great similarities among each other. The phases 1 and 5 deviate somewhat: the pattern of response 1 is more noisy and not that regular. The response of phase 5 shows again 3 polar winter response zones instead of two, as it was





seen for the phases 1 and 8 in the boreal winter case. Overall, one might in outlines see also here a gradual transition from one MJO phase to the next in a way that the polar structure slowly descents. This time one could start in MJO phase 2 with tracking the polar positive anomaly at $0.01\,\mathrm{hPa}$. It descends down to $0.1\,\mathrm{hPa}$ in phase 4 and further down to about $5\,\mathrm{hPa}$ in the phases

8 and 1. The abrupt change to the next cycle would then be between the phases 1 and 2 (it was between 4 and 5 for the boreal winter / QBO easterly case) and the phase with the additional polar replacement area would be phase 5 (instead of 1 and 8 for boreal winter / QBO easterly). This looks like these characteristic phases are shifted by about half an MJO cycle between the 2 cases, however, one has to consider that we tracked this time the positive anomaly instead of the negative one, which is also a change by half a cycle.

This raises for us the question if maybe the phases with the abrupt changes are only missing for some reason a third polar replacement area in the highest altitudes in both cases, boreal winter and austral winter. If these additional areas were there, one would probably identify a completely smooth transition without any abrupt changes. To exemplify this, one could also start at phase 5 in Fig. 6 with following the upper negative anomaly around $0.01\,\mathrm{hPa}$ with the eyes. This is a start identical to the description for the boreal winter / QBO easterly situation. One can now identify roughly a descend down to $0.1\,\mathrm{hPa}$ in phase

8 and even a further descent from phase 1 to 4, where it is located between 1 and $10\,\mathrm{hPa}$. To close the cycle continuously, one could attribute the lower negative anomaly of MJO phase 5 at $10\,\mathrm{hPa}$ to the end of the descent and the upper one to the start of the new cycle. Hence, one could find a closed cycle for the negative anomaly in the austral winter case. This was impossible for the boreal winter / QBO easterly case because a second negative polar area was missing in the phases 4 and 5 in this case. Hence, this aspect should gain particular attention in future studies with other or longer datasets to clarify if the temperature

response pattern maybe really completes a closed cycle in both hemispheres.

### 3.6 Austral winter / boreal summer and state of the QBO

We have also checked the QBO influence for the austral winter situation and find that the QBO has not such a big impact during this season in contrast to boreal winter situation described before. The results are shown in Fig. 7 for QBO easterly and Fig. 8 for QBO westerly. Although some differences between the MA temperature responses for QBO easterly and westerly are

seen, they do not have a clear systematical structure so that these differences could also be random effects due to the different sampling periods. The only noteworthy difference is that the austral winter / QBO easterly case (Fig. 7) does not show a clear descent of the pattern, whereas this descent can be seen in outlines in the austral winter / QBO westerly case (Fig. 8).

Together with the previous Sect. 3.5, we can conclude that the structure of the responses and the transitions between the phases for austral winter compare best to the boreal winter situation under QBO easterly conditions, while the austral winter

response itself shows no clear QBO influence. Possible reasons for this are discussed in Sect. 4.4.



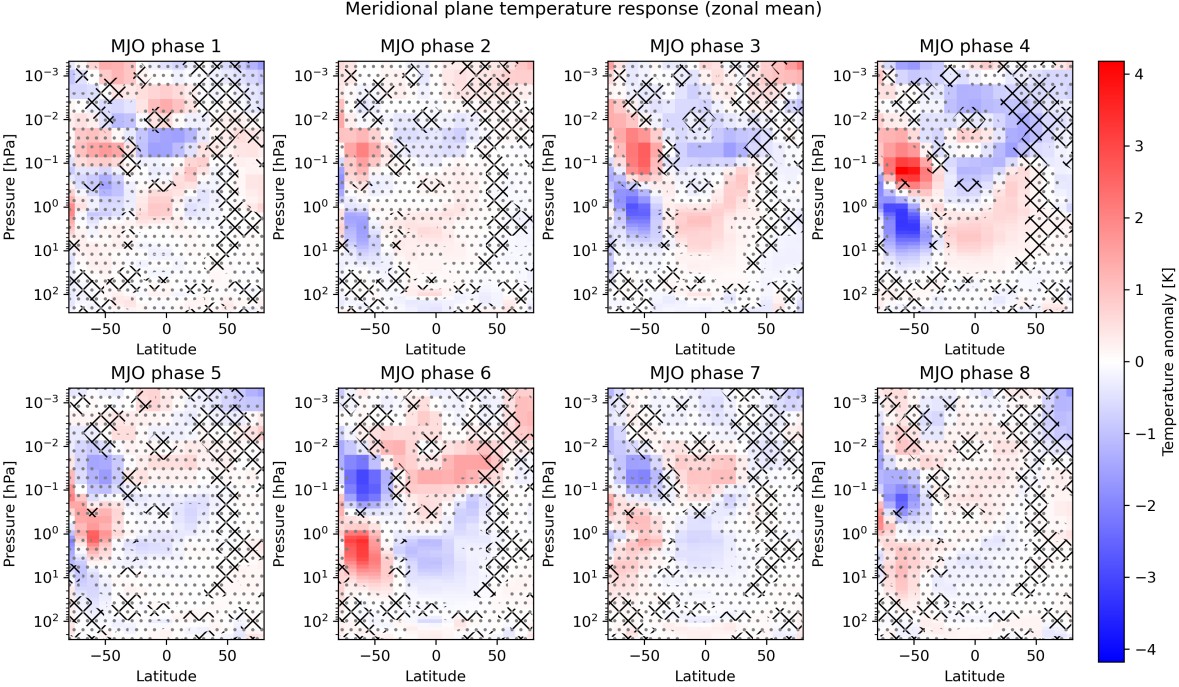

**Figure 7.** Similar to Fig. 6 but for the austral winter / boreal summer and QBO easterly situation. Note that the color scale is different compared to Fig. 6

.

## 4 Discussion

### 4.1 Similarity with interhemispheric coupling

We have carved out a characteristic spatial pattern, the five-zone-response as we called it, of the MA temperature response to the MJO in the meridional plane, which we described in detail in Sect. 3.2 and the following subsections.

It appears that this pattern is similar to the temperature pattern of the interhemispheric coupling (IHC). IHC was first described by Becker et al. (2004) in the context of the anomalous southern hemisphere winter of 2002 and is basically a sequence of dynamical disturbances, which also result in temperatures anomalies. The spatial structure of these anomalies is generally comparable to the one found here. After its discovery, the existence of IHC was further supported by satellite observations of, e.g., noctilucent clouds (Karlsson et al., 2007, 2009b) and simulations with the Canadian Middle Atmosphere Model (CMAM, 395    Karlsson et al., 2009a).

These simulations can be used for a first quantitative comparison of the spatial structure with our results. This comparison reveals that the pattern fits indeed also well in terms of the pressure levels of the strongest deviations (compare, e.g., Fig. 6





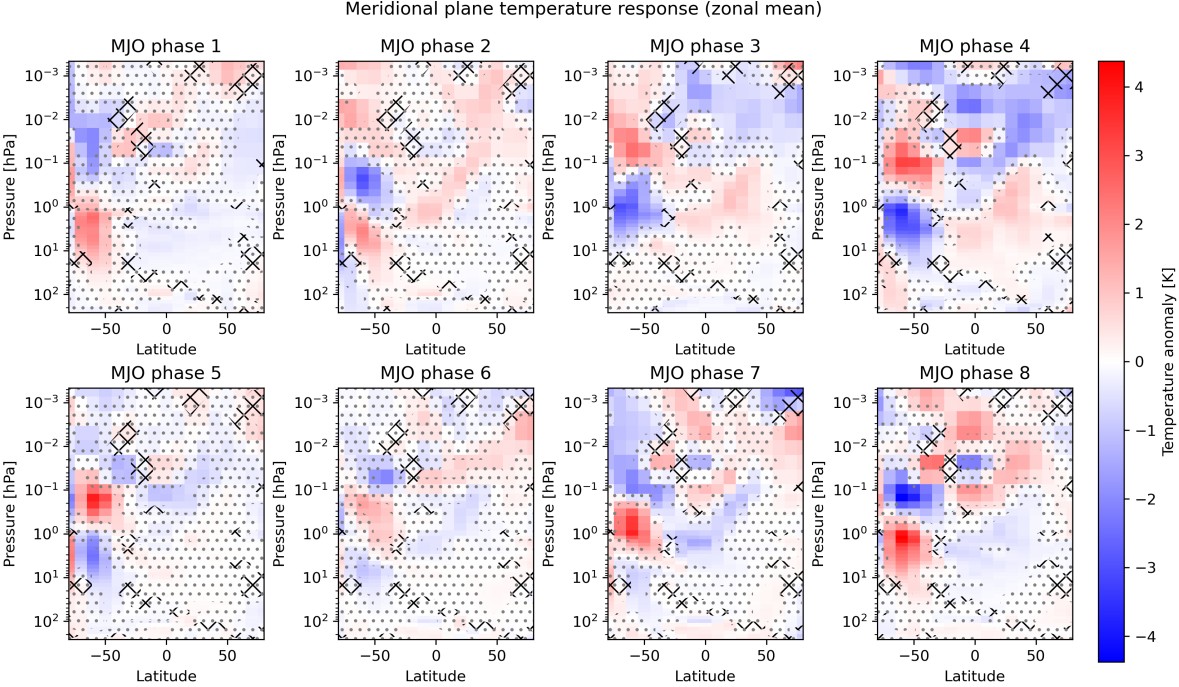

**Figure 8.** Similar to Fig. 6 but for the austral winter / boreal summer and QBO westerly situation. Note that the color scale is different compared to Fig. 6

.

for time lag 15 d in Karlsson et al. (2009a) with our Fig. 2 for MJO phase 5, which are the periods of the strongest responses in both studies). We see, however, a much stronger magnitude of the effect with about $\pm 6$ K for northern winter filtering and

$\pm 10$ K for the northern winter / QBO east filtering instead of $\pm 3$ K in Karlsson et al. (2009a).

Observational temperature anomalies in the context of IHC are reported by Yasui et al. (2021). Although they use the same dataset as in our study, their results may still serve as independent crosscheck because they do not link their results to the MJO at all, so that we will compare the results briefly in the following: Yasui et al. (2021) use a previous version of the MLS data and consider the data only up to 2014, which is, however, not expected to make an important difference. Their focus

is also the boreal winter season, but they do not filter for the QBO. They also compute composites of the MLS temperature data, which are, however, not temporally triggered by the MJO, but by the appearance of SSWs (although more precisely, a correlated equatorial temperature anomaly is used as trigger). Their major result in the present context is the middle column of their Fig. 5, where they show the meridional plane temperature anomaly composites of the MLS data for different time lags around their trigger dates. Generally, they also find the "five-zone-response" pattern, although the summer mesosphere zone is

partly above the highest MLS observation altitude. They see a much stronger magnitude of the anomalies with up to $\pm 30$ K





using the SSW trigger compared to the $\pm 6\,\mathrm{K}$ to $\pm 10\,\mathrm{K}$ we found for the MJO trigger. As the analysis by Yasui et al. (2021) is designed to extract warm polar winter stratosphere situations, they find the five-zone-response only with this sign and the opposite response is not available for comparison to our results. Nevertheless, the similarities of the covered SSW situation with our results of the MJO phases 5 and 6 (extended by MJO phase 7 and maybe 8 if filtering for QBO easterly conditions is

applied) are obvious. The use of a log-pressure coordinate in Yasui et al. (2021) complicates the direct quantitative comparison of the vertical structure with our pressure-resolved results a bit, but ad hoc conversions of the coordinates indicate that the altitudes of at least the stronger polar dipole fit quite well to each other and that also the descent with time is comparable: The dipole descents between the time lags -8 d to 0 d and +8 d to +16 d about $15\,\mathrm{km}$ to $20\,\mathrm{km}$, which roughly fits to the descent from MJO phase 5 to 7 in Fig 4 (although we have applied the additional QBO filtering in this case).

Karlsson et al. (2009a) point out that the same IHC pattern can develop with the opposite sign depending on the sign of the initial dynamical disturbance (planetary wave drag; details on the mechanism follow below). We have seen clearest in Fig. 4 for the boreal winter and QBO easterly filtering that we get the opposite sign of the IHC pattern for MJO phases that are half an MJO cycle (i.e. 4 phases) apart. Specifically, the pattern of MJO phase 6 fits to the stronger wave drag scenario after a time lag of about 15 d in Karlsson et al. (2009a), whereas MJO phase 2 fits to the weaker wave drag scenario with the same time lag

(although phase 3 shows slightly a stronger response). The transition between these opposing structures is further discussed below in Sect. 4.3.

We note that the study by Yasui et al. (2021) contains two more details in their temperature anomalies and their schematic of the mechanisms (their Figs. 5 and 15), which might be comparable to mentioned features of our results. We have not checked the quantitative similarity in terms of the vertical levels etc., but still we would like to briefly mention these possibly common

features: first, Yasui et al. (2021) have not only considered a vertical dipole as the temperature response above the winter pole, but consider a third anomaly zone above the mesopause, which has the same sign as the stratospheric anomaly. We have seen this in the responses of individual MJO phases, e.g., phase 8 for the boreal winter and QBO easterly situation (Fig. 4). Second, Yasui et al. (2021) partly show that the summer mesospheric response of the IHC may itself be a vertical dipole with a positive and a negative anomaly for certain time lags, which we have found at least for the austral winter situation (Fig. 6).

Another subtle but noteworthy detail appears when comparing our seasonally resolved results to Fig. 2 of Karlsson et al. (2007) although the quantities shown are not directly comparable: Without going into the details of their analysis, one important indicator for the IHC is the blue area in the northern polar area in the top panel of their figure. The top panel resembles our boreal winter situation and it is seen that the blue area is directly located above the winter pole. When considering the austral summer situation (their middle panel), the blue area is expected above the southern hemisphere polar latitudes. There is indeed

a blue area, but it is not located directly above the pole but instead shifted northwards to about -50° latitude. We see a similar difference when comparing the polar temperature responses of our analysis. Whereas the polar response dipole is directly located above the northern polar region during boreal winter (Fig. 2), it is shifted somewhat more towards the equator for the austral winter situation (Fig. 6).

Overall, we conclude from these comparisons that the temperature response pattern found here in relation to the MJO

resembles the one that is discussed as the characteristic structure of IHC.





A mechanism for the IHC was already proposed in the first publications and then detailed in Körnich and Becker (2010). The major points are outlined here only briefly with the notice that details, which are not very important for our study, are still subject of the scientific discussion and refinement as, e.g., in Yasui et al. (2021). An overview is given in Fig. 1 of Körnich and Becker (2010). The starting point is a deviation of the planetary wave (PW) drag in the winter stratosphere. The theory

of IHC is at that point independent of the origin of this deviation. The changed PW drag changes the background dynamical conditions in the winter stratosphere, i.e. the strength of the westerlies and the Brewer-Dobson circulation. This changes also the adiabatic vertical motions and influences thereby the temperatures in the winter stratosphere above the winter pole and close to the equator. This explains the lower two temperature deviations of the quadrupole. The changed stratospheric circulation also changes the filtering of vertically propagating gravity waves, which influences the mesospheric meridional circulation, which

has again a direct influence on temperature via adiabatic vertical motions. Altogether, this explains the upper two temperature deviations of the quadrupole. The propagation of the signal to the summer mesosphere, where the fifth temperature anomaly is located, is then explained by the changed equator to summer pole temperature gradient in the mesosphere and corresponding dynamical anomalies.

The results presented in Sect. 3 and discussed here now suggest that the MJO can be a source for the initial PW drag

anomaly, which triggers the IHC anomaly pattern. On the one hand this might be expected, since the MJO is known to interact with PWs (e.g., Lau and Waliser, 2012, Chapter 14), which conveys its influence to the extratropics (e.g., Cassou, 2008) and the stratosphere (e.g., Garfinkel et al., 2012; Yang et al., 2017, 2019). On the other hand it could also be expected that the signal is washed out on its way to the mesosphere considering that the MJO evolves simultaneously further in the troposphere and is probably involved in further interactions with PWs.

We note that we are not aware of any explicit description of a potential descent of the IHC pattern in the theory, as we have found it here. Intuitively, one could speculate on two possible initial reasons for such a descent: First, the descent could be an inherent feature of the IHC mechanism, so that an IHC pattern that is once triggered will always descend in the time after. Second, it could be a specialty of the MJO-IHC interaction in a way that each MJO phase triggers the IHC pattern at a different altitude. In any case, having the decent of the pattern in mind could foster further research on IHC in terms of theory and

observations: A possible decent should be considered in theoretical studies for a further refinement of the IHC mechanism and some observational studies are based on evaluating data at only particular altitude levels, which might miss some of the effect due to the varying altitude levels of the pattern.

## 4.2 Connection to SSWs and planetary wave forcing

The IHC pattern is connected to SSWs, since SSWs may produce or amplify the polar winter part of the IHC temperature

pattern. Garfinkel et al. (2012) has statistically linked the occurrence of SSWs to the evolution of the MJO phases. We have now statistically linked the IHC to the MJO. Hence, statistical connections between all the three features have been made and the linkages should be mutually consistent, at least in so far as the strong variability allows for rough comparisons. This appears to be the case, as we will briefly discuss in the following.





We have shown above (Sect. 4.1) that the MA temperature responses to MJO phases 5 and 6 (extended by 7 and 8 in case

of the QBO easterly filtering) during boreal winter fit to the IHC coupling pattern shown in Yasui et al. (2021). The situation analyzed in Yasui et al. (2021) is basically a SSW situation, so that roughly the second half of the MJO cycle (phases 5 to 8) can be attributed to an SSW-like pattern, whereas the first half of the cycle shows a the pattern with reversed sign.

The results of the second half of the MJO cycle can therefore be compared to those of Garfinkel et al. (2012), where the MJO-SSW connection is directly explored. Garfinkel et al. (2012) find MJO phase 7 as the dominating phase directly

preceding SSWs (time lag of 1 to 12 days). This phase is also part of the second half of the MJO cycle, so that the results are at least consistent in this respect. Considering that the analyses differ in their setups (Garfinkel et al. (2012) consider the extended boreal winter, do not distinguish between the two QBO states, use a different MJO index, etc.) and that these analyses are always complicated by the inherent large variability, it might not be reasonable to discuss quantitative similarities and discrepancies in detail. However, an additional semi-quantitative observation can still be made: From our analyses, one would

have concluded that MJO phases 5 and 6 (instead of phase 7 highlighted by Garfinkel et al. (2012)) show the strongest SSW-like responses (compare Figs. 2 and 4). But one has to keep in mind that the pattern in our analysis descents from phase 5 to phase 8 at least for the boreal winter/QBO easterly case, so that we see a continuous evolution of the pattern. Garfinkel et al. (2012) use instead a sharp SSW trigger, which is based on the atmospheric state at $10\,\mathrm{hPa}$, which is at the lower bound of the polar positive temperature anomaly found here for MJO phases 5 and 6, but more central in the positive anomaly for phase 7.

Hence, what we already identify as a SSW-like pattern in phases 5 and 6 might with regard to the exact altitudes be a precursor of the real SSW-like pattern, which is then in terms of altitudes only reached in phase 7. This would also be roughly consistent with the result of Garfinkel et al. (2012) that MJO phase 6 is a precursor for SSWs with a longer time lag of 13 to 24 days. As said before, these comparisons should only be treated as rough qualitative cross checks and not as reliable quantitative results.

The joint reason of SSW and IHC occurrences is a deviation in the planetary wave forcing of the winter stratosphere. Whereas

the forcing is always stronger than normal for SSW situations, the IHC coupling pattern has also been found for the opposite case. Karlsson et al. (2009a) call those two IHC modes strong or weak planetary wave events, respectively. As expected, the IHC pattern for the strong events resembles that of SSWs. The findings of the present study put now the MJO in the context of these deviations: The MJO acts probably as one source of such planetary wave disturbances. Moreover, it can both weaken and strengthen the planetary wave activity and the previous discussion suggests that the first half of the MJO cycle (phases 1

to 4) tends to weaken the planetary wave activity, whereas the second half (phases 5 to 8) tends to strengthen it.

The effect of MJO phase occurrences on wave activity in the northern hemisphere winter stratosphere is explored in more detail by Wang et al. (2018a). They find that a higher occurrence frequency of MJO phase 4 is in line with weaker wave activity and a stronger polar vortex in the stratosphere, i.e. the opposite of a SSW situation. This is roughly consistent with our findings in the sense that MJO phase 4 belongs to the first half of the MJO cycle, for which we also found the opposite of the

SSW situation. The reason is according to Wang et al. (2018a) not that the MJO induced wave activity is generally weaker, but that MJO wave activity is in antiphase with the climatologically apparent waves in the respective region for MJO phase 4. Furthermore, Wang et al. (2018a) state that the opposite is true for MJO phase 7, so that a higher occurrence frequency of phase 7 results in higher wave activity and a SSW-like situation. This is also roughly consistent with our findings, since




phase 7 belongs to the second half of the MJO cycle. However, Wang et al. (2018a) state that all other MJO phases should not have a strong influence on the extratropical stratosphere, because there is no clear in-phase or antiphase relationship of the MJO related waves and the climatological waves. We find instead a response throughout the middle atmosphere for roughly all MJO phases. This would only be consistent if the clear MA responses found here for all MJO phases were time-lagged results of the forcings of only the MJO phases 4 and 7. In this case it would, however, be dubious that the main forcing occurs towards the end of those consecutive MJO phases, which lead to the same MA response according to our results (phases 1 to 4 and 5 to 8). Hence, the explanation of Wang et al. (2018a) for all phases except 4 and 7 seems not to be in line with our findings and it appears appropriate to question if Wang et al. (2018a) really describe the full mechanism. We note also here that the statistical approaches are not directly comparable. Whereas we make a composite analysis including data of all days with a certain MJO phase, Wang et al. (2018a) analyze changes that are correlated with changes in the occurrence frequency per winter of individual MJO phases and, hence, effects that are more relevant to the interannual time scale.

## 4.3 Timing of the MA response with respect to the phase transition of the MJO

### 4.3.1 General remarks

We have described before that there are characteristic responses of the MA temperature for each MJO phase with systematic transitions between them. The coarsest classification of the eight individual responses would be the discrimination of the two cases "weak wave driving" and the SSW-like "strong wave driving". The two classes consist roughly of the subsequent MJO phases 1 to 4 (weak) and 5 to 8 (strong), particularly in the boreal winter and QBO easterly case (Sect. 3.3). Wang et al. (2018a) provide a possible explanation for such two classes, although it appears to us that it does not completely explain our results, as discussed in Sect. 4.2.

Particularly, the responses also evolve gradually within these two classes, which manifests itself in a descent of the spatial structure from phase to phase. This results in an almost continuous cyclic development of the MA temperature responses over the eight MJO phases (details and limitations of this generalized statement are extensively discussed in Sect. 3, particularly in the Sects. 3.3 and 3.5).

Indeed, it appears plausible that a periodically evolving trigger, like the MJO, should also produce roughly a systematically and periodically evolving response as we have found here. However, in terms of mechanisms it appears quite remarkable to us that the response evolves so systematically. This is not only because the MJO is only quasi-periodic with strong variability and hemispheric and seasonal asymmetries. Moreover, the MJO propagates itself zonally and changes its properties gradually while evolving in time. At each point in time, the current state of the MJO could in principle interact with other relevant parts of the atmosphere, causing disturbances, which travel horizontally and vertically into the MA, where they could in principle interfere with arbitrary other disturbances of the preceding MJO states. This could lead to a great variety of superimpositions of MJO responses in the MA, which could also, e.g., totally cancel out each other or look much noisier than the results found here. Instead, the overall net effect of the MJO seems to be more like a simpler periodic trigger that initiates the IHC pattern





with the one or the other sign on a coarse scale and that causes, seen on a finer scale, gradual altitude shifts from phase to phase.

The clarification of the exact mechanism is not within the scope of this paper. Contrarily, in the light of many-faceted properties of the MJO-MA temperature interaction brought up in Sect. 3, we think that coming to a full explanation needs

future research dedicated to this topic.

However, independent of an explicit formulation of the mechanism, the statistical relationship between the MJO phases and the MA temperature, which is described in this study, might already serve as a benchmark for complex atmospheric models. Particularly those models, which are used for intraseasonal weather forecasting depend on having a good representation of the MJO and its teleconnection patterns in which the stratosphere is involved. We have carved out a characteristic and complex

response pattern for the major parameter temperature, which connects the MJO with the complete MA in all latitudes and heights and we think that this pattern and its evolution with the MJO phases can be used as a challenge for atmospheric models, which implicitly concerns many aspects of atmospheric couplings. In order to support these kinds of evaluations, we briefly interrelate our results with previous studies in the following subsection 4.3.2.

### 4.3.2    Relation to previous studies

There are a few studies, which go beyond the specific case of the MJO-SSW connection discussed before (Sect. 4.2) and give more general information on the MJO-MA temperature relationship for particular MJO phases, which can be compared to our results. However, strict comparisons are difficult, not only because the analysis setups and datasets differ, but also because most studies analyze the state of the MA only after selected MJO phases while considering relatively long time lags. Hence, they study the temporal evolution after a certain state of the MJO. This is certainly useful to explore causalities of the propagation

mechanisms of the disturbances and to examine lead times of potential predictions based on the MJO states. But it also bears ambiguities because the MJO evolves itself during these relatively long time lags. E.g., a state of the MA that is found 30 days after MJO phase one, could also be attributed to MJO phase 4 or 5 with almost no time lag assuming one MJO phase lasts for about 7 days. We have intentionally decided to study the statistical relationships without time lags, but with respect to all eight MJO phases as alternative an approach, although we might extend these analyses to a treatment of time lags in the future for

comparison purposes. For now, comparisons have to ad hoc assign the MJO phase evolution to specific time lags, which is of course difficult due to the high variability of the MJO. As an example, also Garfinkel et al. (2012) mention such an assignment, which considers a periodicity of 30 to 60 days for a complete MJO cycle and thus must remain a rough estimate.

The influence of the MJO on mesospheric temperatures during boreal winter has recently been analyzed by Sun et al. (2021). They partly also use the MLS dataset employed here, but mostly rely on modeled data of the model SD-WACCM. A period

of 100 days around SSWs is excluded, which further limits the comparability to our results. Sun et al. (2021) concentrate mainly on the temporal evolution after MJO phase 4 and on the northern hemisphere. As one main result they report a cooling of the northern hemisphere mesosphere 35 days after MJO phase 4. In qualitative agreement, we also see the mesospheric cooling after MJO phase 4, namely in the phases 5 to 7 (Fig. 2). However, converted to time lags this appears to be earlier than the time lag of 35 days. Sun et al. (2021) also show the meridional plane temperature response of the MA for different



time lags after MJO phase 4 (their Fig. 2). In consistency with our results, also the polar vertical dipole is seen with the change of the sign at about $1\,\mathrm{hPa}$. Furthermore, they also show the temporal switch of the dipole's sign, here between time lag $25\,\mathrm{d}$ and $30\,\mathrm{d}$. They do not show much of the equatorial area, but a reversed equatorial dipole can be perceived to some extent. The order of magnitude of the temperature anomaly is with $\pm5\,\mathrm{K}$ comparable to our analysis, as well as the fact that the strongest anomalies are observed, when the stratosphere shows the warm anomaly. The agreement of the orders of magnitude

is particularly remarkable, since the periods of strongest variability are missing due to the exclusion of SSWs. Overall, we conclude that apart from the fact that differences are expected due to the different approaches, the general response pattern is roughly consistent with our findings, while the timing is hard to compare, so that consistency in this respect cannot be evaluated.

A study broadly comparable to Sun et al. (2021), but focused on the stratosphere and based on reanalysis data is presented

by Yang et al. (2019). They analyze the northern hemisphere stratospheric temperature response to partly all MJO phases, but highlight a stratospheric warming after MJO phase 4 with a time lag of $30\,\mathrm{d}$. This can be regarded as consistent with our results in the same sense as discussed before in the context of Sun et al. (2021) for the mesosphere. Furthermore, the order of magnitude of the temperature response is in their analyses comparable to our results of the boreal winter situation without QBO filtering (Fig. 2), i.e., in the situation for which the analysis setups are closest to each other. Their Fig. 1(a) shows the

evolution of the response with the MJO phases at $10\,\mathrm{hPa}$ for a polar cap average. Comparing their results for a time lag of $0\,\mathrm{d}$ with our polar results reveals a general consistency in the sense that the stratospheric response is negative for MJO phases 2 and 3 and positive for the phases 5 and 6, as in our study. Altitude-resolved results are also shown in their Fig. 2 (left column) and should be visually compared by the readers themselves.

Yang et al. (2017) present a similar analysis based on reanalysis and modeled data, but with respect to the austral winters

and, thus, roughly comparable to our Fig. 6. In an overview of the southern polar cap temperature response to all MJO phases at $10\,\mathrm{hPa}$ (their Fig. 2 (A) and (B)) they find cold anomalies at a time lag of $0\,\mathrm{d}$ roughly for the MJO phases 1 to 3 and warm anomalies for the phases 5 to 7. This is roughly consistent with our results; however, we see the more pronounced corresponding responses shifted by one MJO phase, namely 2 to 4 for the cold anomaly and 6 to 8 for the warm anomaly. Yang et al. (2017) discuss the responses to the MJO phases 1 and 5 in more detail and present data for the complete meridional

plane for these phases. The temperature response for MJO phase 5 (their Fig. 4) appears roughly consistent with our results. Particularly, the results for a moderate time lag of $10\,\mathrm{d}$ after phase 5 fit quite well to our results of following MJO phase 6. Also, our magnitude of the anomalies, which is smaller than during boreal winter in the northern hemisphere, fits roughly to the results of Yang et al. (2017). Note that this plot in Yang et al. (2017) also contains a summer mesosphere response in the northern hemisphere, which also fits to our results. Our results for MJO phase 1 are quite noisy and hard to compare.

A study by Yoo et al. (2012) is actually focused on the tropospheric connection between the MJO and the Arctic, however, the upper vertical range of their analysis slightly overlaps with the lower edge of our analysis around $200\,\mathrm{hPa}$, so that a rough comparison is possible. Still, one should have in mind that the datasets could be affected by boundary effects etc. in these locations especially because these pressure levels are not in the focus of the respective analyses. Yoo et al. (2012) use reanalysis data and concentrate on the MJO phases 1 and 5 during boreal winter. Their Fig. 6 shows meridional plane



temperature responses for different time lags, of which the upper parts can be compared to the lower parts of our Fig. 2. Their response for MJO phase 1 with time lag 0 d shows a positive anomaly over the Arctic, which is consistent with the Arctic response at lower vertical levels in our analysis. A time lag of 8 d might roughly be attributed to MJO phase 2 in our analysis and also here positive anomalies in this region are found in both studies. Also their response for phase 5 with time lag 0 shows a positive response in higher levels of the Arctic region, which is also roughly consistent with our analysis. Interestingly, the sign of the response changes in their study when looking at time lag 8 d after MJO phase 5, which might be attributed roughly to MJO phase 6 in our study. Consistently, we also find a change to a negative anomaly in this region with the transition from phase 5 to phase 6. Also the order of magnitude of the response is with about $\pm 1\,\mathrm{K}$ consistent between the studies. As stated before, we are cautions with the interpretation of this small overlapping region. However, if the similarities were accepted as real, this could mean that the responses at the lower pressure levels in our study could be more a result of the tropospheric processes analyzed by Yoo et al. (2012) rather than a result of stratospheric processes discussed here so that they do not have to be explained within the dynamical framework of IHC. We also note that Yoo et al. (2012) report on stronger poleward Rossby wave propagation in connection with MJO phase 5 and weaker poleward wave propagation in connection with phase 1. They do not discuss a potential vertical propagation into the stratosphere, so that their findings are not necessarily applicable to our case. Nevertheless, this attribution is roughly consistent with our finding that the MA response to the first half of the MJO cycle resembles the weak wave driving IHC pattern and the second half the strong wave driving IHC pattern (Sect. 4.1).

### 4.4 Influence of the QBO

As shown above, filtering the data not only for the boreal winter but additionally for QBO easterly conditions, carves out the IHC pattern more clearly in terms of stronger anomalies and smoother transitions of the pattern between the MJO phases (Sect. 3.3). For this influence on the MJO-MA connection, at least two possible types of interference with the QBO have to be considered: The influence of the QBO on the MJO itself as well as the inner-stratospheric influence of the QBO on the polar stratospheric dynamics.

### 4.4.1 Influence of the QBO on the MJO

The influence of the QBO on the MJO is currently an active field of research. Yoo and Son (2016) found that the MJO amplitude indeed varies with the QBO phase during boreal winter in a way that the MJO is stronger during the QBO easterly phase and weaker during the QBO westerly phase. Afterwards Zhang and Zhang (2018) argued that the MJO is not necessarily stronger during the QBO easterly phase, but that the number of active MJO days (days during which a strength threshold is exceeded) is increased. Wang and Wang (2021) state that both kinds of interpretations are actually connected and should not be treated independently. In any case, the fact that there is something like a strengthening of MJO effects for the QBO easterly phase during boreal winter fits qualitatively very well to our findings that the MJO effect on the MA temperature is most pronounced under these conditions. Note that this would be a net effect of the tropical lower stratosphere on the complete MA conveyed by the troposphere, where the MJO is active.



The QBO influence on the MJO has mainly been reported for boreal winter. More precisely, Yoo and Son (2016) and Son et al. (2017) state that they find no significant relationship of the QBO and the MJO for other seasons. However, Densmore et al. (2019) claim that there is also an QBO-MJO relationship during boreal summer, which is reversed compared to boreal winter. While the analysis by Densmore et al. (2019) is limited to the MJO phases 4 and 5, it also applies two more sophisticated analysis approaches, which might help to identify a potentially weaker signal also during boreal summer: First, the state of the QBO is determined more sophisticatedly and, second, an BSISO index is used to characterize the state of the MJO during boreal summer in addition the common MJO indices.

Our examination of the QBO influence on the MJO-MA connection during boreal summer (Sect. 3.6 and Figs. 7 and 8) showed no clear indication for a QBO influence during boreal summer. This can be seen as consistent with both previous notions: We cannot identify a strong impact of the QBO during boreal summer like Yoo and Son (2016) and Son et al. (2017), but we also do not use the more sophisticated QBO identification approach and the specialized BSISO index as in Densmore et al. (2019), which would support the identification of weaker signals. Hence, our study remains inconclusive with respect to the QBO influence during boreal summer, but a more sophisticated future analysis might be of interest, which could, as a side note, also benefit from an extension of the analysis by Densmore et al. (2019) to the other MJO phases.

Overall, there is obviously a seasonal asymmetry of the QBO influence on the MA temperature response to the MJO (Sects. 3.3 and 3.6): The QBO easterly phase is needed to produce the clearest response pattern during boreal winter, whereas a comparable (although weaker and more noisy) structure is produced during boreal summer in the southern hemisphere regardless of the QBO phase. One possible explanation for this apparent asymmetry could be the generally higher variability of the northern hemisphere boreal winter situation. This might demand for a particularly strong MJO signal to dominate the intraseasonal variability in the northern hemisphere. And the influence of the QBO easterly phase on the MJO during boreal winter could be a major factor to facilitate this strong MJO signal during this season. In contrast, a weaker MJO might already be able to dominate the signal in the more quiescent southern hemisphere, so that a QBO support is, regardless of its still debated existence, not needed to produce the austral winter signal. Another explanation (which strictly speaking belongs to the next Sect. 4.4.2), could be the modification of the occurrence rates of SSWs by the QBO, which is roughly only relevant during boreal winter, because SSWs only rarely happen at all in the southern hemisphere.

### 4.4.2 Influence of the QBO on the polar stratosphere

The second possible type of interference is the inner-stratospheric influence of the equatorial QBO on the polar winter stratosphere. This connection was first described by Holton and Tan (1980) and implies in the current context of temperature that the polar winter stratosphere tends to be colder during QBO westerly phase and warmer during the QBO easterly phase (e.g., Labitzke and van Loon, 1992). We note that this observation is connected to a known influence of the QBO on the occurrence of SSWs during boreal winter (e.g., Camp and Tung, 2007), so that this aspect is not independent from the discussion in Sect. 4.2. This could mean that, e.g., the strength with which the MJO can influence the polar stratospheric temperature by modulating SSWs is itself affected or limited by the QBO's influence on SSWs.




In any case, the QBO is overall known to have a similar effect on the polar stratosphere as was found for the MJO in this
study, but on the biennial rather then on the intraseasonal time scale. One could expect intuitively that these two effects appear
superimposed in the current analysis, so that particular combinations of MJO phases and QBO states interfere constructively
or destructively and therefore lead to extreme values of the stratospheric temperature anomaly. As described before (Sect. 3.4),
the QBO westerly response for boreal winter in MA temperature is not as systematic and less significant as the QBO easterly

response for the same season, so that it is almost impossible to deduce a deeper systematic behavior and we cannot examine
this intuitive expectation in detail. However, we found that the polar winter stratospheric responses for some MJO phases are
actually reversed for the QBO easterly and westerly situation, e.g., the cold polar stratospheric responses for the QBO westerly
phase appear partly during the same MJO phases (e.g., 6 and 7) as the warm responses for the QBO easterly phase and not
during the opposite MJO phases (Sect. 3.4 and Figs. 4 and 5). This does not seem to be consistent with this simple expectation

and suggests a more complex connection.

We note that together with the QBO influence, also the solar 11-year cycle has been shown to be important for the polar
winter stratospheric temperature (e.g., Labitzke and van Loon, 1992). This is worth mentioning because of two reasons: First,
the solar 11-year cycle is another source of variability, which is not considered in this study and which could therefore disturb
the pure signal of MJO and QBO in MA temperature. The influence of the solar variability should therefore be analyzed in

a future study. Such a study would also be of interest, because an influence of solar variability directly on the MJO has also
been indicated (Hood, 2017; Hoffmann and von Savigny, 2019), so that there are also at least 2 types of interference possible.
Second, also the influences of QBO and solar 11-year cycle on the polar stratospheric temperature turned out to be not additive
(e.g., Labitzke and van Loon, 1992; Camp and Tung, 2007), similar to what our analysis indicates here for the superimposed
QBO and MJO influences.

Overall, we cannot discriminate the relative importance of the two possible types of QBO interference with the MJO influ-
ence on MA temperature introduced here and possible further influences like the solar cycle. Whereas a consistency of major
aspects can be seen for the QBO influence on the MJO itself (Sect. 4.4.1), we cannot draw any reliable conclusions on the
importance of the inner-stratospheric influence discussed in this section. This might be possible in the future, when longer
datasets allow for a more detailed filtering with respect to other environmental parameters like the solar activity.

The discussion of the inner-stratospheric QBO influence on the MA temperature response to the MJO in this section was
so far restricted to the polar winter stratospheric temperature. This region acts as the starting point for the emergence of the
IHC pattern (Sect. 4.1) and it has indeed been shown that the QBO influence on this region also influences other regions of
the MA according to the IHC mechanism (Espy et al., 2011). Moreover, Murphy et al. (2012) show that some details of IHC
mechanism themselves are modified by the QBO phase. Hence, also the temperature response to the MJO in the complete MA,

which is conveyed by the IHC mechanism, as we have indicated before, should be subject to this QBO influence. Therefore,
the complete MA can and should be considered to further isolate the relevant effects in future studies.



## 5    Summary and conclusions

There is rising interest in MJO influences on the middle atmosphere: First, because the stratosphere is one route for MJO-related teleconnections, which is, e.g., of practical interest for weather forecasting efforts on the intraseasonal timescale. Second, it is of interest for MA research to identify all sources of variability in the MA, e.g., in order to cleanly isolate anthropogenic signals. And third it is naturally of interest for basic research on troposphere-stratosphere coupling. The MA temperature is obviously a key parameter in this context. Contrarily, only few studies deal explicitly with the influence of the MJO on MA temperature. The available studies are typically mostly based on modeled or reanalyzed data and are limited to individual MJO phases or particular geographical regions.

We aimed at contributing to this field by statistically analyzing the connection of the MJO and the MA temperature based purely on observations. We included, moreover, all MJO phases in the analysis, in order to reveal also the transitions of the MA response when the MJO progresses from one phase to the next. As statistical approach, we have applied a superposed epoch analysis to the 17-year global temperature dataset of the MLS satellite instrument. For the MJO characterization, we have used the OMI MJO index. We have distinguished different environmental conditions, particularly different seasons and states of the QBO. For the sake of brevity, we have focused this paper on zonally averaged data.

We have indeed found that the MA temperature is influenced by the state of MJO in large areas of the MA and under roughly all considered atmospheric conditions. Still, the strength of the influence varies considerably with both the atmospheric conditions and the region of the MA.

Aside from more occasional weaker spatial response areas, a pronounced characteristic response pattern, which we called here "five-zone-response", manifested itself under many atmospheric conditions. The first two zones (compare. Fig. 3) are two anomaly areas with opposing sign in the polar winter stratosphere and mesosphere, together constituting a dipole response above the winter pole. These zones show mostly the strongest anomalies among all five zones. A second dipole, comprising the third and forth zone, is found in the equatorial stratosphere and mesosphere. This dipole has a reversed sign compared to the polar winter dipole. The fifth zone is found above the summer pole in the mesopause region, so that the five-zone-response spans almost the complete meridional plane. The anomaly of the fifth zone has the same sign as the equatorial mesospheric zone. The overall sign of the complete five-zone-response is different for different MJO phases. It has been checked with two Monte Carlo approaches that the anomalies of at least these five zones are broadly significant for most analyzed cases.

A pattern like the five-zone-response in known in MA research from the interhemispheric coupling mechanism. This IHC mechanism basically describes, how an initial disturbance in planetary wave drag in the winter stratosphere propagates due to a chain of different dynamical effects through the MA until it reaches the summer mesosphere. It thereby causes a similar pattern of temperature disturbances as we have seen here for the MJO influence. More detailed comparisons of the particular locations of the disturbances with the original IHC publications have confirmed the similarity. Moreover, the IHC mechanism works for weak and strong planetary wave activity, causing the one or the other sign of the temperature anomaly pattern as we found it here for different MJO phases.





Hence, one major outcome of the present study is the finding that the MJO can lead to the initial planetary wave drag
disturbance. The MJO can therefore systematically trigger the IHC mechanism and it can do this with the one or the other sign,
i.e. with a strengthening or weakening of the planetary wave driving in the winter stratosphere. The present study therefore
connects an atmospheric feature mostly known in tropospheric research, the MJO, with one mostly known in MA research, the
IHC.

The analysis, applied individually to boreal or austral winter data, has revealed that the five-zone-response appears in both
seasons with the response being roughly mirrored at the equator. The boreal winter response is, however, much stronger.
Furthermore, the response during the boreal winter season shows also a strong dependence on the QBO phase, which we have
not convincingly seen for the austral winter. Overall, the strongest anomalies and the clearest spatial pattern appear for boreal
winter combined with the QBO easterly phase. In this case, we have found temperature anomalies in the range of $\pm 10\,\mathrm{K}$

with the strongest anomalies being located over the winter pole. For boreal winter and the QBO westerly phase the pattern is
much less clear but the anomalies are still in the range of $\pm 6\,\mathrm{K}$. For austral winter the anomalies are with $\pm 3\,\mathrm{K}$ much smaller,
but nevertheless, the spatial pattern can easily be identified for most MJO phases. When the austral winter data is separated
according to the QBO phases, the anomalies are with $\pm 4\,\mathrm{K}$ slightly higher, but the spatial pattern is by far not as sensitive to
the QBO as during boreal winter.

One possible reason for the strong QBO influence during boreal winter is a potential alteration of the MJO strength by the
QBO during this season. This QBO-MJO connection has been proposed in the literature in the last few years. This kind of
connection would indeed be consistent with our findings such that the easterly phase of the QBO strengthens the MJO during
boreal winter, as we have discussed above in more detail. This would mean that the QBO influences the tropospheric MJO,
which afterwards influences the MA temperature with a corresponding strength.

A dynamical feature in the MA closely related to the IHC are SSWs, which are mostly observed in the northern winter
hemisphere apart from only two exceptions that occurred in the southern hemisphere so far. They essentially produce a strong
positive temperature anomaly in the polar winter stratosphere, which is then part of the IHC temperature anomaly pattern.
During the last decade, an influence of the MJO on the occurrence of SSWs has been reported in the literature. Since we have in
this study now linked the MJO also to IHC, we have checked and confirmed that there is generally a mutual consistency of these

three features in terms of the MJO phases, for which a warming in the polar stratosphere is found. We note, however, that the
strong atmospheric variability as well as different analysis setups and properties of the datasets only allow for rough consistency
checks instead of precise comparisons. Therefore, the readers should pay particular attention to the detailed discussion above
if more multifaceted information is needed.

Two more aspects related to SSWs are here of interest. First, the fact that SSWs occur mostly in the northern winter hemi-

sphere, where they produce large temperature deviations in the polar regions, fits well to our finding that the largest temperature
anomalies of our complete analysis occur in this area. This could be seen as support of the idea that the MJO and SSWs are
interconnected and produce together the strongest responses correlated to the MJO phases. However, one could also argue that
possible uncorrelated temperature deviations caused by the SSWs are so large that they are not completely eliminated by our
statistical approach, so that the polar winter response is only (partly) an artifact. While we cannot totally rule out the latter, we





have discussed several arguments, which indicate that the boreal winter polar temperature anomalies are really correlated with the MJO and probably amplified by the occurrence of SSWs. One of these arguments is that the same pattern of the temperature responses to the MJO can be seen in the southern hemisphere winter, even if the periods with the rare SSWs in this hemisphere are excluded from the analysis. The second aspect of interest in the context of SSWs is that a connection between the QBO phase and the occurrence of SSWs is known such that the QBO easterly phase supports the occurrence of SSWs. Together with
the MJO-SSW connection, this could be a second reason for the strong QBO dependence of the MA temperature response to the MJO during boreal winter.

We also analyzed the change of the MA temperature responses while the MJO transitions through its eight phases. For most atmospheric conditions, we could identify a systematic behavior, which is again most clear for boreal winter and QBO easterly conditions. It can be described in two steps: First, the responses to the eight phases can mostly be attributed to either a
"weak wave driving" or a "strong wave driving" class according to the sign of the five-zone-response. Moreover, responses that are attributed to the same class belong to subsequent MJO phases (e.g., for the boreal winter and QBO easterly situation the responses to the MJO phases 1 to 4 belong to the weak wave driving class whereas the responses to the phases 5 to 8 belong to the strong wave driving class). This supports the idea that we observe a real systematic connection between MJO an MA temperature with our statistical approach. Second, for many transitions from one MJO phase to the next a gradual shift of the
five-zone-response to lower altitudes can be observed, suggesting the MA reacts systematically to each of the MJO phases. We are not aware of any explicit discussion of such a descent of a temperature anomaly pattern in the literature, which is related to the IHC or the MJO influence on MA temperature. Hence, this is likely a new aspect, for which a mechanism has still to be established. E.g., it appears not to be directly clear whether each phase of the MJO triggers the IHC pattern at a different altitude or if the IHC pattern inherently descends after it has be triggered once by a certain MJO phase or a combination of
both.

Overall, we have presented statistical evidence for a connection of the tropospheric feature MJO and the MA feature IHC and have supported with additional arguments that a real causal connection likely exists. We think that this is a noteworthy example for the complex couplings across different atmospheric layers and geographical regions in the atmosphere: in this case the source of disturbances is located at lower altitudes in the tropics (while it is probably itself influenced from above) and turns
out to have observable implications up to 100 km altitude and from pole to pole. Moreover, the results further demonstrate the interconnections of different atmospheric features, particularly the MJO, the IHC, the QBO, and SSWs.

Because of the wide coverage of atmospheric regions and included dynamical features, we think that the observational results presented here can help to further constrain the underlying mechanisms. For the same reasons and because of the simplicity of the resulting temperature response pattern, the results presented here appear also to be a good benchmark for atmospheric mod-
els. This applies particularly for those models, which depend especially on a good representation of troposphere-stratosphere couplings on the intraseasonal timescale, like, e.g., models that are used for intraseasonal weather predictions.



*Code and data availability.* The releases of the OMI calculation package are freely available from a persistent storage referenced by Hoffmann (2021), while ongoing improvements are continuously found at https://github.com/cghoffmann/mjoindices, last access August 16, 2022. The MLS temperature data is available for download at https://doi.org/10.5067/Aura/MLS/DATA2520, last access September 15, 2022.
NOAA Interpolated Outgoing Longwave Radiation (OLR) data provided by the NOAA PSL, Boulder, Colorado, USA, is available from their Web site https://psl.noaa.gov/data/gridded/data.olrcdr.interp.html, last access August 16, 2022. The zonal wind data for the QBO characterization is provided by Freie Universität Berlin at their website https://www.geo.fu-berlin.de/en/met/ag/strat/produkte/qbo/index.html, last access August 16, 2022.

*Author contributions.* CGH has outlined the project, performed the analysis and written the manuscript. LGB has carried out a pilot study
under the supervision of CGH and CvS showing the potential of the analysis, CvS has extensively discussed the approach and the results during all phases of the project. All authors have reviewed, discussed and improved the manuscript.

*Competing interests.* The contact author has declared that neither they nor their co-authors have any competing interests.

*Acknowledgements.* This work was supported by the University of Greifswald. We are grateful to the teams of the NASA MLS instrument, of the NOAA OLR dataset, and of the QBO dataset maintained by the Freie Universität Berlin for providing their valuable datasets. The
analysis has benefited from discussions on natural variability of the MA temperature and MA dynamics within the DFG (German Research Foundation) research unit VolImpact, particularly within the subproject VolDyn.



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
