# Peer review of "Signatures of the Madden-Julian Oscillation in Middle Atmosphere zonal mean Temperature: Triggering the Interhemispheric Coupling pattern"

_EGUsphere, 2022_

## Author Comment (AC1)

**Authors final comments to the public discussion on egusphere-2022-998**

We would like to thank both reviewers very much for the taking time to go through our manuscript and writing the constructive reviews with many suggestions for improvement. We would also like to apologize for submitting our responses late.

Before replying to each of the reviewers' comments individually, we would like to start with a general comment, since many of the individual criticized aspects seem to have a common basis.

**Content**

**General comment to both Reviews**

In our perception, many points of both reviewers' criticism can be connected to the following three points

1. The physical mechanism for the described statistical relationship is not analyzed, explained or precisely discussed. Furthermore, the statistical method is relatively simple and other methods could be applied to support the findings.
2. The discussion of the connection to the previous studies (Section 4) is too long.
3. The new aspects of the manuscript are not clear.

We generally understand that the manuscript can be criticized in these aspects (particularly the first two aspects) and understand the reviewers' concerns on one side.

However, we would like to mention that these apparent deficiencies are the result of an intentionally chosen focus of the paper, which differs from previous publications and was maybe not described clearly enough. This focus should be complementary to existing studies, which follow more the structure that is proposed by the reviewers. We would like to make this clearer in the following.

Many of the studies, which we relate our results to, have indeed a greater mixture of presenting new atmospheric findings (e.g., temperature signals) and working out the underlying mechanisms. For this purpose, the studies also have to mix different kinds of datasets (observations, reanalyzes, atmospheric models), atmospheric variables (temperature, wind, wave drag) and partly also statistical or numerical methods. While these papers contribute indeed more to the explanation of the mechanisms, these papers also have to stay focused, which is mostly achieved by reducing the number of geographical and atmospheric conditions (concentration on restricted geographical areas and atmospheric altitudes, selected seasons, individual MJO phases, states of the QBO etc.), which makes it partly difficult to mutually compare the results. Hence, these studies present mechanisms for selected cases, but in our view the overarching picture remains somewhat patchy.

Our idea is therefore complementary. Coming from satellite data analysis, we aim at reducing the complexity of combining different datasets, variables and methods in the presentation to allow for a broader overview in terms of atmospheric regions and conditions. We achieve this by focusing descriptively on a single aspect, the statistical connection of middle atmosphere (MA) temperature and MJO phases. This results in a self-contained and closed description of what is statistically contained in this global satellite dataset w.r.t to the MJO influence. In other words, we intentionally restricted our work to a "what-can-be-found-in-this-dataset"-perspective. In our view, this approach has some benefits, which support existing and future studies on the underlying mechanisms:

a. The analysis is completely based on a single observational dataset and therefore requires less assumptions on the reliability and representativeness of the used datasets.

b. The simplicity of the used statistical method allows for a simple reproduction. In particular, it is easy to check if models are able to reproduce the found patterns.

c. Due to its overview-character, the study could help to relate the existing studies to each other and hence help to overcome the patchiness. One aim of the longer Section 4 was already to work in this direction.

d. Seeing the broader geographical overview of at least the statistical relationship helps to identify a broader overall picture. Indeed, we could recognize the interhemispheric coupling (IHC) pattern in the MA temperature response, which, to our knowledge, is a new finding.

e. We think that the characteristics of our study outlined here so far can help to initiate future investigations of the mechanisms of the MJO-MA coupling by explaining this broader picture and reproducing its emergence in atmospheric models.

While we agree that individual parts of the results are also contained in previous studies (and are investigated there in more detail), one new achievement of our analysis is that it describes a common observational frame for these results. A second new achievement is that this frame could be linked to the known MA feature, the IHC.

We also agree with the reviewers that results based on a purely statistical analysis should at least be supported by showing a physical plausibility. However, our approach in this respect is not to investigate the mechanisms ourselves in this paper. Instead, we discuss the links to previous studies, demonstrate the mutual consistency where possible, and with that make their explanations of the mechanisms applicable to our results. This is the second reason for the comprehensive discussion in Section 4.

We also think that the additional analysis steps proposed by the reviewers are of interest to further understand the origin of the statistical relationship. In fact, we have some of them in mind for future studies (see individual responses for details). However, we think that integrating these analyses into the present paper (which is already somewhat lengthy as the reviewers note), would go beyond the scope of the manuscript with the particular focus outlined here.

Overall, it remains somewhat unclear to us, to which extent the manuscript has to be revised. We certainly learn from the reviewers' criticism that we have not made our intention and focus of the paper clear enough. Hence, we will definitely work on an improvement of the presentation, for which we will of course consider the reviewers' suggestions in this respect. However, we are not sure if the reviewers still require us to change the scientific focus of the publication or if they can support

the general idea of the paper in the light of our clarifications here. Therefore, we will keep such changes to a minimum for the time being and will explain our decisions point by point below.

**Response to Reviewer 1**

**Note**: We repeat the reviewer's statements in grey, while our responses are typeset in black. If we recognize several aspects of criticism in one comment, we will separate our answer in individual bullet points. Intended modifications of the manuscript are highlighted in **bold** face.

Reviewer 1: This paper investigates the statistical temperature feature in the middle atmosphere with respect to the Madden-Julian Oscillation, considering the interconnections of seasonal variation, the QBO, and SSW based on the analysis of the satellite observation. The authors suggested that the middle atmospheric temperature response to the MJO under different atmosphere conditions manifests an interhemispheric coupling with a "five-zone" pattern. The comprehensive comparisons show that this temperature response to the MJO agrees with those suggested by previous works. The result may benefit the models' coupling process within the intra-seasonal timescale. The issue addressed in this study is well within the journal's scope. However, the analysis and discussions presented in this study lack robust logic and verification. I would suggest major revisions before it is accepted for publication.

**Response**: We thank Reviewer 1 very much for taking the time to review the manuscript and appreciate the suggestions, which we will consider to improve the manuscript. We note that some of the suggestions are formulated relatively general, so that it is unfortunately somewhat difficult for us to find possible modifications, which precisely address the criticized aspects of the paper.

Reviewer 1:

Major comments:

1. At a fundamental level, the analysis results in this article are consistent with the findings of previous studies, but the new findings are unclear. In my opinion, the results obtained from the analysis in this paper may be different from the existing studies about the MJO impact under different phases of the QBO, but an in-depth analysis of the reasons is missing, and only a simple discussion of the possibilities is given. There needs to be more, an in-depth discussion and analysis are necessary.

**Response**:

- "New findings": **We will mention the new findings more explicitly in the revised manuscript**. We have here also tried to make those clearer in the general comment above.

- "Results different under different phases of the QBO": We cannot attribute this very general and somewhat vague statement to particular aspects of our paper, which we could then revise. Does the reviewer mean that the agreement, which we mention for individual aspects, is wrong? Or does the reviewer miss an additional description of disagreements? In any case, going into a deeper discussion in the manuscript would probably cause section 4 to be longer instead of shorter (shortening is generally requested by both reviewers).

- "in-depth analysis of the reasons is missing": We are not sure if this statement applies to the possible discrepancies w.r.t the QBO phases (see point above) or to the findings of the paper as a whole. In any case, it is unfortunately also not clear to us, what the reviewer would actually consider to be an in-depth discussion or an in-depth analysis, but we guess that this refers to elucidating the physical mechanisms of the statistical relationships. As written in the general comment, we wanted to give the paper a different, more descriptive, focus in combination with links to aspects in previous studies, where mechanisms are partly already explained.

Reviewer 1:

2. When analyzing the boreal winter response of atmospheric temperature to the MJO, the authors do not differentiate the effect of whether or not SSW events were included in the results. In particular, the authors recognize that the strong perturbation caused by SSWs cannot be effectively disentangled from the weak perturbation caused by the MJO itself. Thus, the MJO effect on atmospheric temperature during the boreal winter in this paper is of limited importance for related studies.

**Response**:

- In general: It is true that we have not shown any analysis, which only considers boreal winters without SSWs. We have mentioned this ourselves in the manuscript. We have nevertheless performed a quick experiment, which actually only considers winters without SSW (as we have mentioned in the manuscript on page 11) and in which the signal is still seen. To support this statement, we include a respective preliminary figure (Figure 1 at the end of the document). We have decided not to include details of this quick experiment in the paper, because it is less robust than the other presented results due to only limited

remaining days in the data. Instead, we have discussed further indications for the existence of the signal in the absence of SSWs in the manuscript (page 10).

- "effect on atmospheric temperature during the boreal winter in this paper is of limited importance for related studies": We agree that one aspect is actually missing in the paper due to the reason of limited data: a quantitative estimation of the boreal winter temperature response to the MJO in the absence of SSWs. I.e., a figure, which is similar to what we have shown for the other cases, particularly for the boreal winter with and without SSW together. However, we do not agree that this limits generally the importance of all the boreal winter results in the paper because of two reasons: Firstly, as mentioned in the manuscript and shown by Garfinkel et al., 2012, SSWs are influenced by the MJO themselves. Furthermore, SSWs are also part of the IHC pattern when they occur. Hence, SSWs are part of the explanation of the temperature responses to the MJO and should not be eliminated except for the one particular research question, which considers exactly the winters without SSW (which is of course also interesting). Secondly, SSWs are not rare events on the northern hemisphere, but are approximately as probable as quiet winters. Hence, also related future studies would probably compare the SSW case and not only the SSW-free case.

Reviewer 1:

3. It has been shown in previous studies that it takes some time for the MJO to affect the polar stratosphere via planetary waves, but the analysis presented in this paper is based on a synthetic analysis with no time delay. The results obtained may, therefore, not capture true MJO effects. The temperature response seen in phase 5 is a superposition of delayed phase 2 and phase 4 effects. Therefore, it may be possible to make the results more convincing if the results of different statistical approaches could be demonstrated.

**Response**:

- In general: We agree, that a similar study but resolved into time-lags after the individual MJO phases would be of interest. Such analyses are often seen in related studies, which focus more on the mechanisms. In fact, we have already prepared a first version of such an analysis. However, we have decided not to include it in the manuscript, because we had the impression that the additional discussion of a temporal dimension could make the paper even longer and with that harder to understand for the reader. We plan to include a final version of those results in a future publication. However, as discussed in the manuscript (page 23), also those time-lagged analyses bear ambiguities because the MJO evolves itself

during the time lags. And since the MJO period is so variable, it is also here unclear, which MJO states are exactly averaged for certain time-lags.

- "The results obtained may, therefore, not capture true MJO effects": We think that this depends on the kind of interpretation. We agree that due to the duration of the propagation of the signals into the MA, we can indeed not claim that a particular MJO phase is the causal reason of the temperature pattern, which we find for that MJO phase. Instead, from a causal perspective, it appears also for us more likely that the previous MJO phases may also have an influence and even more that a shown temperature pattern is caused by a mixture of the influences of several MJO phases. But we can claim based on our analysis that the temperature anomaly pattern, which we find for a particular MJO phase, is the statistical mean state at the time of the given MJO phase; and this is actually our main statement. We think that this statistical claim is (although being clearly weaker than a causal one) still useful. This applies, e.g., for comparisons with other publications, which also use this kind of analysis or, e.g., for model comparisons, which should be able to reproduce the interfering causal influences of the individual MJO phases. By the way, we also stated in the manuscript that these inferences are of importance, e.g., for an explanation of the altitude shift of the pattern with the evolution of the MJO phases. But we think that a detailed disentanglement of the influences of the individual phases is beyond the scope of the paper. **In any case, we will make the difference between the causal and the statistical interpretation clearer in the revised manuscript.**

- "superposition of delayed phase 2 and phase 4 effects": We unfortunately do not comprehend this particular example in detail and leave it with our general remark.

- "if the results of different statistical approaches could be demonstrated": We don't know which kind of approaches the reviewer has actually in mind. We would think of a time-lagged analysis, which is in our view too extensive for the present manuscript (see above).

Reviewer 1:

4. The structure of this paper consists primarily of the analysis of the MJO composite under different atmospheric conditions and a detailed comparison to the results of previous studies. As such, this paper appears to be a mixture of a simple statistical analysis of the data and a review of MJO effects on the middle and upper atmosphere rather than a research paper addressing a specific scientific question. To give the reader a better understanding of the main content of this article, I would suggest that the article should be revised, the logic of the article should be rearranged, and the focus of the article should be emphasized on the new findings.

**Response**: As stated in the general comment, the structure of the paper followed a certain idea. But we understand that this idea is not clearly obvious from the written text, **so that we will certainly work on an improvement of the structure and a shortening of the discussion**. Nevertheless, it is difficult to estimate for us from the comment, which kind and extent of restructuring could actually convince the reviewer, **so that we will try to strike a balance between the original idea and a clearer structure**.

**Response to Reviewer 2**

**Note**: We repeat the reviewer's statements in grey, while our responses are typeset in black. If we recognize several aspects of criticism in one comment, we will separate our answer in individual bullet points. Intended modifications of the manuscript are highlighted in **bold** face.

This paper studies the response of the middle atmosphere (MA, i.e. 261 hPa – 0.00046 hPa) to the MJO using daily MLS temperature for the period of 2004 to 2021. The MJO signature is studied using composite analysis for the eight MJO phases. The same analysis is repeated for easterly and westerly QBO phases and then for summer and winter months. I have some major concerns about the analyses which are stated below.

**Response**: We thank also Reviewer 2 very much for taking the time to review the manuscript and appreciate the partly detailed suggestions, which we will consider to improve the manuscript.

Reviewer 2:

**Major comments:**

- Physically speaking, high-frequency variability of stratospheric polar vortex is known to be controlled by wave drag in association with planetary and/or gravity waves. Figure 1a and Figure 4 suggest that the polar vortex has a MJO signature that is marked by cold anomalies in the earlier phases of the MJO but warm anomalies in the later phases. I believe that at least one of below is required: 1) Wavelet coherence analysis shows that daily MJO index and daily zonal mean temperature at 80N, 3hP are significantly coherent with each other and the MJO leads the temperature at 80N, 3hPa. See "A Practical Guide to Wavelet Analysis", C. Torrence and G. P. Compo, 1998 for detail. Some equivalent analysis to show that the polar vortex and the MJO share the same variability would also be helpful;  2)  Significant anomalies in wave drag for the MJO phases that show significant temperature anomalies in the polar region;  3) The MJO leads temperature anomalies in the MA or the SSWs. In fact, 3) has been studied by Garfinkel et al. (2012) in terms of MJO influences on the SSWs. 2) is not easy to perform using temperature because wave drag or EP flux divergence require other parameters. 1) should be feasible to perform using temperature data.

**Response**:

- "I believe that at least one of below is required": We thank the reviewer 2 particularly for these thoughts on 3 ideas to further support the findings and also for directly estimating their applicability. We discuss those in the following in the order of applicability estimated from our perspective:
  - Idea 3 ("The MJO leads temperature anomalies in the MA or the SSWs"): As the reviewer writes himself, the idea 3 has already be investigated in terms of SSWs by Garfinkel et al., 2012. This is actually our basic line of argument in the manuscript: In Section 4.2 of the manuscript, we show a basic consistency of the results by Garfinkel et al., 2012, and our analysis, so that the reasoning of Garfinkel et al., 2012, can broadly be applied to our results (particularly since we do not distinguish between boreal winters with and without SSW, see above). We also refer to their results w.r.t to the leading role of MJO phase 7. Hence, from our perspective the reviewer's requirement "I believe that at least one of below is required" is already fulfilled in the manuscript via idea 3.
  - Idea 1: "Wavelet coherence analysis": We also think that this method can be more directly applied to our data than idea 2, however the application appears still to be non-trivial due to the data gaps caused by the filtering of the data by the seasons, the MJO strength and the QBO state. We agree that is method could be generally of interest in a future study, which is more focused on the underlying periodicities etc. However, we are not sure if the added information to the present manuscript really justifies the effort, as we will try to justify in the following. In our understanding the reviewer asks us to demonstrate two aspects: Firstly that MJO and MA temperature vary "significantly coherent" and secondly that "the MJO leads the temperature". The composite analysis applied by us is actually a method particularly made to show the first aspect, since only co-variations will not be eliminated by the averaging. Hence, we think that the wavelet transform analysis will lead to redundant results for this aspect. The second aspect is indeed not covered by our analysis, but indirectly checked by the link to the study of Garfinkel et al., 2012, as outlined in the point above.
  - Option 2 ("Significant anomalies in wave drag"): As the reviewer writes himself, this kind of analysis is beyond the scope of our data analysis and the manuscript. Also here, references in the manuscript to previous studies, which also include such analyses, might help to give the reader an impression of these processes. Still, we agree that this kind of analysis is of interest for a future study.

In summary, we think that the reviewer's requirement is already fulfilled in the manuscript via idea 3. The other two ideas are definitively of interest for future studies with a somewhat different focus, but are either beyond the scope of the present analysis or would not add much independent information (compared to the effort) to it. **In any case, we will improve the text to make the implications of the study by Garfinkel et al., 2012, more obvious in the text and therefore to explicitly mention idea 3.**

- The much-strengthened MJO-MA temperature connection could be physically real. But the authors need to make the mechanism clearer. For instance, how does the MJO affect the jet streams over the North Pacific whereby it affects the strength of planetary waves from the troposphere? Firstly, the strength or frequency of the MJO is enhanced during easterly QBO. Diabatic latent heating associated with extensive MJO convection can generate Rossby waves which propagate poleward and eastward toward North America, altering the atmospheric circulation as well as the planetary waves that propagate upward into the winter stratosphere. As it stands, this chain of effects is not clear either in the analysis or in the lengthy discussion.

**Response**: Yes, it is true that the mechanism is not independently analyzed or discussed in the paper. Please notice the general comment in the beginning of this document for an explanation of our idea of the focus of the paper. Since the reviewer mentions already key aspects of a mechanism, it appears straight-forward to integrate a description of such a chain in the paper. However, our impression is that if such a description is included in the paper, it should be accompanied by respective analyses of data, which support each step. This would require a much more comprehensive analysis (see also previous point), which would be beyond the scope of this paper.

**Reviewer 2:**

- If I understand the method correctly, the temperature anomalies globally are effectively band-pass filtered to a temporal width of 10-90days and a latitudinal band width of 10 degree. It is thus incorrect to state that the composite differences shown are MA response to the MJO. Even if the differences are statistically significant, they can be also interpreted as the co-signal of something else and/or be interpreted MA impact on the MJO. This is because the analysis perform does not ensure that the MJO leads the zonal mean signals in the MA.

For instance, the so-call interhemispheric coupling could force both the MJO and the polar vortex. Hood (2017;2018) also showed both solar and QBO can influence the MJO.

**Response**: Yes, we agree that the direction of the influence is not determined or even a joint reaction to a third process is strictly not excluded by our analysis, which is usually the case for correlation analyses and which also applies for the statistical approach used here. Also in response to reviewer 1, **we will make the statistical nature of the analysis clearer in the revised manuscript and point the reader to these limitations**. That said, as discussed in the points above, there are strong indications in the publications, which are referenced in Section 4, that there is actually an influence from the MJO in the direction of the MA temperature. One example is the already mentioned study by Garfinkel et al., 2012. Another indication is that the mechanism of the IHC starts in the stratosphere with a disturbance from below. However, we firstly agree that the physical connection should be shown in future and secondly we do not exclude that the process rebounds to the troposphere and particularly the MJO. **We will integrate a statement on this in the revised manuscript.**

Reviewer 2:

- I wonder how many of the samples in DJF QBOe subgroups are serially correlated. For instance, how many daily samples are adjacent to each other in time or how many daily samples belong to the same MJO event? The question applies for DJF QBOw and JJA QBOe etc. This is because the 10-day running average applied to the temperature data. Only one sample within the 10-day window should be regarded as independent statistically.

**Response**: This is a good point and we agree that this is not clear from the paper.

- "Statistical dependence": We use the 10-day filtering primarily for reasons of comparability with other analyses. We have repeated the analysis presented here without applying the 10-day moving average and the results do essentially not change. We have included respective sample figures (Figure 2 to Figure 4 at the end of the document). Hence, the statistical dependence, which we strictly introduce ourselves, does not affect the results. **We will add a respective explanation in the revised manuscript.**
- "how many daily samples belong to the same MJO event": We use every available day and not only, e.g., the starting days of each MJO phase. We think that this is justified (considering the statement made in the previous point w.r.t the statistical dependence), since the temperature signal can vary on each individual day, so that consecutive days with a constant MJO phase could still produce different temperature signals. The number

of days, which go into the analysis when filtered for MJO strength (always the case), season and QBO state is roughly 50, as we have also stated in the manuscript on page 6.

**Reviewer 2:**

- Is it possibility to detect MJO signature according to the QBO phases without contamination from many other factors, such as ENSO, QBO, solar and volcanic eruptions in this case? Given MLS data set only covers the period of 2004-2021, during which there has been only ~7.3 QBO cycles on average. Together with the band-pass filter applied, it could be very hard to interpret the results.

**Response**: We agree that it is difficult to unambiguously separate the QBO influence from other comparatively slowly varying influences if those showed (by chance) a correlation to the QBO during the analyzed period. We have mentioned this with respect to the solar 11-year variation ourselves in the manuscript (page 27), but **we will add a more general warning in the revised manuscript**. We have performed some more ad-hoc tests with additional filters. However, the results are difficult to interpret due to the lower sample size and we decided to leave those results for a future publication, in which either the database is more robust or a stronger focus can be laid on the statistical details. We still decided to present the QBO analysis for reasons of comparability, since many other studies also combine a seasonal and a QBO filtering in the first place before introducing additional filters**, but we will make the limitations clearer in the revised manuscript.**

**Reviewer 2:**

**Specific comments:**

- Abstract requires to be shortened. Bring out the key results that are closely relevant to the tittle of the paper. It would be helpful if the authors can get the line "a major outcome of the present study is the finding that the tropospheric MJO can trigger the IHC mechanism, which affects many areas of the MA" sooner and start to explain the exact areas of the MA are affected by the MJO via the IHC mechanism.

**Response**: **We will revise the abstract and consider the reviewers suggestions.**

**Reviewer 2:**

- Could the weaker MJO-MA temperature connection be due to QBO disruption? E.g. A westerly phase of the QBO was disrupted in 2015/16.

**Response**: This idea of an influence of the QBO disruption sounds generally interesting, but unfortunately, we have difficulties in understanding the given example precisely. Could it be that some words are missing in the comment? Which weaker "MJO-MA temperature connection" is actually meant?

**Reviewer 2:**

- Section 4 is far too long. It could be very helpful if it is a review paper. But they do not really help the readers to better understand the results presented in this paper. The discussion should be shortened and focus on the key results obtained.

**Response**: As outlined in the general comment, section 4 was intended to be an alternative way of supporting the plausibility of the statistical results obtained in this study by closely linking them to the literature. **In any case, we will revise section 4 and try to shorten it as requested while also making the intention clearer and keeping it useful w.r.t to its original purpose.**

**Reviewer 2:**

- In several places, the authors stated that the MJO signal in the MA depends on atmosphere conditions. But what exactly does the "atmospheric conditions" mean? The MJO itself would be one of atmospheric conditions. Please define the term properly.

**Response**: We meant the set of atmospheric properties, which we use for the filtering of the data, i.e., the season, the state of the QBO, and the MJO strength. **We will make this clearer in the revised manuscript.**

**Reviewer 2:**

- In the abstract, it mentioned that "the complex couplings across different atmospheric layers and geographical regions in the atmosphere". It would be better to put this sentence in Conclusion section where the authors can be more specific about the complex couplings regarding the MJO influences with concrete results support such a statement.

**Response:** We stated in the abstract "It is therefore a noteworthy example for the complex couplings… ", because we learned from some feedback from the community that this can actually be

an interesting point for other researchers. Hence, we believe that particularly some researchers might try to identify papers dealing with complex couplings and it would be helpful to have such a hint in the abstract. **Hence, for now we would prefer keeping a comment like this in the abstract. However, since we have to revise the abstract anyway, we will reconsider this when we have the new context clearly in mind**. In any case, mentioning this is the abstract does motivate being more specific about this elsewhere in the manuscript, so that this does not appear to be a contradiction to us.

**Response to the Community Comments CC1 to CC3 by Paul Pukite**

**Note**: We repeat the comments in grey, while our responses are typeset in black.

**CC1**: One of the phases of MJO is shifted fom the high-resolution southern oscillation index (SOI) by 21 days.

soi vs mjo

So the signatures of the two are the same, only that MJO becomes a travelling wave left in the wake of ENSO

Enso ship

**CC2**: Brain dead interface -- here are the figures

**CC3**: Apparently, can't just copy a link

https://imagizer.imageshack.com/img921/7305/bXNFwm.png

https://imagizer.imageshack.com/img923/8939/lzIRem.png

**Response**: Thank you very much for your interest in our preprint. We followed the links to the figures and considered your comment, but cannot identify a direct implication for our manuscript. Our manuscript shows a statistical connection of the tropospheric signal of the MJO (quantified by the index OMI) and variations of the temperature in the middle atmosphere. For this, the exact cause of the MJO itself and also the quantitative characterization of the MJO (i.e. which index is used) is not of high importance. If there are other indices describing the same tropospheric signal, the analysis could also be based on those indices. These indices would be expected to be correlated with the MJO index OMI, so that the expectation for a repeated analysis would also be that broadly similar results will be obtained. Possibly these results would show a phase shift depending on the definition of this particular index.

**Additional figures**

[Figure]

*Figure 1: Analysis of the boreal winter/QBO easterly condition but with SSW periods roughly excluded. To achieve the exclusion, we have used the SSW dates of the NOAA SSW compendium (https://csl.noaa.gov/groups/csl8/sswcompendium/majorevents.html) and have excluded all days in the range from -25d to +50d around the SSW dates from the analysis. Major outcome is that the IHC pattern is still visible for many MJO phases (best seen for MJO phase 6). This ad-hoc analysis should be considered as absolutely preliminary.*

[Figure]

*Figure 2: Repetition of Fig. 4 in the manuscript (boreal winter, QBO easterly) but without the 10-day filtering.*

[Figure]

*Figure 3: Repetition of Fig. 5 in the manuscript (boreal winter, QBO westerly) but without the 10-day filtering.*

[Figure]

*Figure 4: Repetition of Fig. 7 in the manuscript (austral winter, QBO easterly) but without the 10-day filtering.*

**References**

Garfinkel, C. I., Feldstein, S. B., Waugh, D. W., Yoo, C., & Lee, S. (2012). Observed connection between

stratospheric sudden warmings and the Madden-Julian Oscillation. *Geophysical Research*

*Letters*, *39*(18), L18807. https://doi.org/10.1029/2012GL053144

---

## Author Response (AR2)

**Responses to the reviewers' comments.**

**Note**: We repeat the reviewers' statements in grey, while our responses are typeset in black.

Changes made in the manuscript are highlighted in red.

**Reviewer 1**

Following my previous comments, the authors revised the article, improved the description of the motivation for the study, and revised the discussion of the findings in the current version of the article. I am appreciated that the authors further prove the robustness of the statistical connection between the MJO and MA temperatures by showing the results in different cases, such as whether bandpass filtering is performed or whether SSW is considered. At the same time, the authors have revised the mechanisms by which the MJO may affect the middle and upper atmospheric temperatures and have discussed and speculated in more detail the possible roles of the MJO, SSW, QBO, and other atmospheric phenomena in Section 4. I only have some minor suggestions:

**Response**: We would like to thank reviewer 1 for taking the time to go through the manuscript again and review the changes we made. We appreciate that the reviewer has acknowledged the improvements of the manuscript achieved with the previous revision.

1 The MA response mentioned in the abstract can reach 10 K, but such a temperature response is mainly for the mid and high latitudes of the winter hemisphere, while the effect on the summer hemisphere via the IHC is much less significant, which needs to be clarified.

**Response:** This information was removed while shortening the abstract in the last interation, but we have now again added a respective sentence.

2 As the authors mentioned, the MJO affects MA probably via the action of planetary waves. A simple analysis of planetary wave activity based on the observed temperature would be helpful, even if this paper does not intend to do a detailed analysis of the mechanism. The analysis of planetary wave activity can be very helpful in understanding the regions and processes occurring in the IHC.

**Response**: Also after careful consideration, we are not really sure, how a satisfying *simple* planetary wave analysis could exactly be conducted: It should not add much content to the paper, but at the same time it should be robust enough to remove the reviewer's doubts which are apparently raised by the present analysis. However, the reviewer classifies the points raised here as "minor" himself and there has no strict requirement for the present manuscript been formulated by the reviewer. Therefore, we treat this point as suggestion for our future work, which must then not be restricted to a *simple* planetary wave analysis. We would like to point out again, that we generally agree that the presented statistical connection should be further investigated also w.r.t to the underlying physics.

3 I also agree with another reviewer that we should not be limited to the composite analysis approach. As suggested by another reviewer, wave coherence should help understand the MJO-related MA IHC activity.

**Response**: As written in the last iteration, we also agree that further analysis approaches are of interest in the future, but still think that they would not fit very well into the present manuscript, which is still

required to be shortened with this revision. As before, we treat this point as suggestion for our future work, since the reviewer classifies himself the points raised here as "minor" and as there has no strict requirement for the present manuscript been formulated by the reviewer. By the way, many points raised by the reviewers in both iterations will serve as guidelines for our future work and we would like to thank both reviewers once more for the detailed comments.

**Reviewer 2**

This paper studies the response of the middle atmosphere (MA, i.e. 261 hPa – 0.00046 hPa) to the MJO using daily MLS temperature for the period of 2004 to 2021. The MJO signature is studied using composite analysis for all the eight MJO phases. The same analysis is repeated for easterly and westerly QBO phases and for boreal and austral winters / summers. The paper the paper has benefited by considering of previous reviewers' comments. The authors have done a relatively good job this round as it becomes easier to follow the motivation, the methods and the results. But I find the discussion section is unnecessarily long and hard to follow. I recommend the authors to consider the comments below before the paper is accepted for publication.

**Response**: We would like to thank also reviewer 2 for taking the time again to go through the manuscript and for the very detailed comments w.r.t to content and language, which we appreciate. It is obvious that the reviewer favors a different, probably much shorter, presentation style. While we see also the advantages of papers, which are optimized for information density, we also find that those are partly harder to read and often harder to reproduce. We chose the longer style intentionally for this manuscript, which possibly connects two different areas of atmospheric research, but will still try to further optimize it in the sense of the reviewer's comments by including as many suggestions as appear compatible with the original style. We will specify this below in the individual answers.

Main comments:

1) The results shown in figure 6 are most interesting and possibly most robust as well. Although the magnitudes of the MA temperature anomalies are smaller than those in boreal winter, the statistical significance of those MJO signals in Austral winter is evidently higher, thus more believable in terms of the so-called "interhemispheric coupling". The magnitude of the signal is also agreeable with existing literature, e.g. Karlsson et al. (2009a).

**Response**: We thank the reviewer for sharing this view on the results. We agree that the interpretation of the austral winter results appears to be clearer due to the weaker (maybe correlated and uncorrelated) variability during this season and area. We are not sure if this comment is meant to indicate a requirement for a change in the manuscript. And since the specific interest in individual parts of the results is probably different from reader to reader, we decided to make no (unrequested?) change in the manuscript. Still, we appreciate sharing this thought and we keep this view in mind for future discussions.

2) The MJO signal in MA temperature during boreal winter peaks at the north pole (Figs. 2-5) but during Austral winter, the largest temperature anomalies are found near the polar vortex edge, i.e. at 60S, e.g. Figs. 6-8. This characteristic difference should be mentioned in the text.

**Response**: This is actually already mentioned in the text (lines 437 to 439 in the reviewed manuscript), although it might have become shorter during the first revision. This feature is actually consistent with the IHC literature and therefore part of our reasoning that the description of IHC actually applies to our observations (lines 431 to 437). Still, we have restructured these lines (also in response to the specific comments), so that this feature is easier to notice.

3) The entire paper can be written much more concisely than its current form. There are a lot of repetition and detailed comparison with previous studies. In particularly, Section 4, i.e. the discussion section is currently 7-8 pages. If the detailed comparison is removed, the same message can be delivered in two pages at most. Furthermore, some of the phrases, e.g. "more or less", "roughly", "probably", or "appears" should not be used so frequently. Sentences that are unsupported by the results or only remotely related should be removed to improve the readership.

**Response**: Some aspects of the criticism, particularly w.r.t to section 4, seem to be to some extent contradictory to the judgement of reviewer 1 in his introductory text, where he writes "At the same time, the authors have revised the mechanisms by which the MJO may affect the middle and upper atmospheric temperatures and have discussed and speculated in more detail the possible roles of the MJO, SSW, QBO, and other atmospheric phenomena in Section 4." Hence, the perception of the section seems to be not totally consistent. Still, we can understand many aspects of the criticism of reviewer 2 and have gone through the whole manuscript again to bring it closer to the form recommended by reviewer 2. We will specify this also below in the specific comments. Still, we can imagine that reviewer 2 might have had more drastic changes in mind, but we hope that the new form is now agreeable when considering that the overall style might be seen differently by different people.

4) The authors need to be very careful with their expression in terms of the statistical linkage between the MJO and the MA temperatures. Try to avoid using words such as influence, affect, or response. "Signal" is more appropriate.

**Response**: We have used the word "signal" more often in this revision. However, the other words mentioned by the reviewer do still appear in the manuscript, at least for reasons of variability in the wording. Therefore, we have inserted a sentence at the end of the "Approach"-section (Sect. 2.2), which clarifies the meaning of all of these words.

Specific comments

Lines 5-9, the word "influenced" is too strong as there is no mechanisms revealed by this study. I suggest changing to "We show that the MA temperature anomalies are significantly related to the MJO and its temporal development. The MJO signal in the zonalmean MA temperature is marked by a particular spatial pattern in the MA, which we link to the …. The signal with the largest magnitude is found in the polar MA during boreal winter in the order of ±10 K when the QBO at 50 hPa is in its easterly phase".

**Response**: changed.

Line 11, remove "found".

**Response**: changed.

Lines13-14, remove "Because of the wide coverage … included dynamical features" as it does not add anything to the content.

**Response**: We prefer to leave these sentences in the abstract. We got some feedback from the modeling community that these aspects are actually of interest. Therefore, we think the abstract should contain a respective hint so that the paper can easier be found by researchers, who are interested in atmospheric couplings, but not specifically in IHC etc.

Line 45, "inner-tropospheric connections" -> "internal variability of the troposphere".

**Response**: We wanted actually to refer to the inner-tropospheric connections mentioned in the paragraph before and not generally to "internal variability of the troposphere". We have rephrased our version of the sentence to "than the inner-tropospheric teleconnections mentioned before" to make our intention clearer.

Line 82, remove "key parameter".

**Response**: changed.

Line 89, "consider instead" -> "instead consider"

**Response**: changed.

Line 125, remove "Nevertheless, we might check our boreal summer results with a special BSISO index in future."

**Response**: changed.

Lines 133-135, I wonder how the days are defined for the composite analysis. Are the temperature anomalies are estimated using 90-day or 10-day forward windows? I would recommend using a 91-day or 11-day windows if the temperature anomalies are estimated based on centred averages.

**Response**: We use indeed centered averages and we agree that an uneven window length would be a more symmetric choice. We used the even window lengths since they are more frequently used in the literature and therefore more directly comparable. Having additionally in mind that also window length changes greater than 1 day do not change the results much (as, e.g., stated in the public responses to the initial reviews), we think that a change from 90 to 91 days and from 10 to 11 days would be mainly aesthetically motivated, so that the effort to recompute all the results appears not to be justified.

We have added the word "centered" in the respective sentence.

Lines 141-142, "We take generally only days into account, during which the MJO strength was greater than 1" -> We only consider those days when the strength of the MJO is greater than 1.

**Response**: changed.

Line 143, citations to other MJO related studies are required here. Also remove "apparent on all considered days".

**Response**: Citations inserted; Inserted some details on the common range of used MJO strengths; removed the phrase.

Lines 149-151, Again, it is not clear to me how the days in the MA temperature anomalies are defined, e.g. centred or forward from the day of a given MJO phase?

**Response**: We do not use temporal windows here. Instead, we have the MJO index and the temperature anomaly on a daily grid. Hence, for the epoch averaging, we have a 1-to-1 matching between the state of the MJO and the temperatures at this stage of the analysis.

Line 189, "if a we" -> "if we"

**Response**: changed.

Line 200, "present manuscript" -> "this study"

**Response**: changed.

Lines 201-202, "On the other hand, by considering previous publications, which overlap in particular aspects, it becomes expectable that a physical mechanism actually exists (i.e. that we do not describe a purely statistical artifact) as we will outline in Sect. 4." This does not add anything to the manuscript; consider removing for better flow of the paper.

**Response**: removed.

Lines 204-214, these sentences can be either removed or placed in discussion section. They stop the flow of the paper.

**Response**: We have removed the lines 204-207. We have left the other lines in the manuscript, but moved them to the end of the results section, since we think that they still contain useful information for some of the readers. Readers that are more familiar with the topic can skip this very short subsection based on the subsection title.

Line 217, citations required after "during that season" to support the statement.

**Response**: We did not want to make a scientific statement here with supporting references, but only motivate the order of the subsections. We have rewritten the sentence, so that it sounds less scientifically motivated and added a reference to the discussion section, where all the references are found.

Line 230, "Put in other words, areas …" can be dropped.

**Response**: removed.

Line 232, "whereas these four zones" -> "Whereas these two dipole patterns".

**Response**: changed.

Line 234, "the first four zones" -> "the quadrupole pattern lowerdown"

**Response**: changed.

Line 235: "we will call it temporarily " -> " we denote these MA anomalies temporarily to"

**Response**: changed.

Line 236: "in the following" -> "in sect. 4.1". And remove the sentence after.

**Response**: This change would be incorrect, since we use the term in many more sections and not only in Sect. 4.1. We have left this phrase as it is.

Line 254: "the temperature signals" -> "the MJO signal in the MA zonal-mean temperatures"

**Response**: changed.

Lines 259-260: "It is obvious the temperature anomalies are even stronger than …" -> "It is evident that the temperature anomalies in the boreal winter are of larger magnitude than those shown in Fig. 2."

**Response**: changed, but corrected to in the boreal winter with QBO easterly conditions.

Line 264: "Important is" -> "The importance is"

**Response**: changed.

Line 274: "the negative polar winter anomaly" -> "the cold anomalies in the boreal polar winter"

**Response**: changed.

Lines 290-295, I am completely lost from "However, …" onwards. Consider rephrasing.

**Response:** We have shortened and completely rephrased the passage. Basically, the content of the comment below by reviewer 2 (w.r.t the lines 363-369) was explained in these lines of the manuscript. But we agree that this was difficult to understand and we have used some words of the reviewer's comment below to make it clearer.

Lines 296-305: Condense and move these sentences to Discussion / conclusion section, which would help the follow of the main results.

**Response**: done.

Line 310: I suggest further condensing the discussion in this section or merge it with section 3.3. The five-zone response is only recognizable during MJO phases 5 and 7 (with opposite signed signals to phase 5). There is no clear evidence in terms of signal descent.

**Response**: We have considerably shortened this section and merged it with the above section.

Line 356: remove "(whereas it was …)" to get the main message across better.

**Response**: removed.

Lines 360-363: the gradual descent of the temperature could start in MJP phase 2 in the Austral winter. The signal intensifies during phases 3-4 while they move downwards. Another cold anomaly zone then appears at 40-50S, 0.02-0.07hPa during MJO phase 5 and then gradually descends to the lower levels.

**Response**: Yes, one could also start with phase 2. Of course, one could in principle start with any MJO phase, if it were a really cyclic transition. In any case, we have added a sentence to give the reader the hint that starting with MJO phase 2 lets one see the transition particularly easy.

Lines 363-369: the abrupt pattern reverse shown in Figure 4 between MJO phase 4 and 5 might be just an artefact, e.g. the signal in the MA during phase 4 is rather weak and statistically insignificant.

**Response**: That is true and was already written higher above in the manuscript (lines 290-295), although obviously hard to understand. We have rephrased both paragraphs for more clarity in this respect.

Lines 386-389: "in both directions" -> "in terms of two aspects". Then: 1) to provide a broader picture that integrate and interrelate current and previous studies; 2) to put forward some possible physical explanation in terms of the responsible mechanisms.

**Response**: Changed, although not totally identical with the reviewer's suggestion.

Line 395: be specific about "first publications".

**Response**: done.

Lines 394: It would be very helpful to properly state the mechanism of the IHC if it is identified to be the most important mechanism that is responsible for the MJO signal in the MA temperature. Changes in planetary wave drag is not enough to explain IHC. Gravity wave drag must be involved.

**Response**: We have actually removed a more detailed description of the IHC mechanism in the first revision to meet the criticism by the reviewers that the manuscript reads too much like a review paper. Additionally, since the literature for the IHC mechanism consists of only a few established papers, it is simple for the readers to find the information themselves based on the references (which we have more explicitly specified with the above point), so that we are now ourselves in favor of not repeating the mechanism here. Still, we have mentioned that gravity waves play a role.

Lines 419-439: need to be much condensed to bring out the key message. It is currently too detailed to grasp the main idea that the authors want to deliver.

**Response**: We have shortened and restructured this part. We note that this part already contains the information that reviewer 2 requests in his main comment 2. We have put this information now into the beginning of these lines.

Lines 485-486 and line 491: No evidence provided for the MJO influences on the PW activity in this study. These statements are pure speculation and should either be removed or rewrite as hypothesis.

**Response**: We note that we have started this subsection quite cautiously in order to make the hypothetical character of the criticized statements clear (The title of subsection contains the word

"potential" and the introductory sentence starting in line 483 directly before the criticized lines reads "Although not proven by our statistical approach, the MJO appears to act as a source of the initial PW disturbances"). Nevertheless, we have rephrased the criticized lines to avoid the impression that an active role of the MJO in controlling PWs was proven in our analysis.

Lines 497-505: my quick examination of Fig. 2 signal suggests that the temperature responses in the boreal winter stratosphere is almost entirely agreeable to the finding of Wang et al. (2018a). Note that stratospheric response may involve one month lag in relation to tropospheric wave forcing. I do agree that this work adds additional information in terms of mesospheric responses, which should be emphasised. The apparent downward descent of the high latitude anomalies is just dynamical response of initial wave forcing. Thus, there is no need to make such a lengthening discussion.

**Response**: We have also stated that a general consistency with Wang et al. (2018a) is apparent and only discussed possible gaps in the explanations of Wang et al. (2018a) in the following lines. Overall, we have therefore difficulties in applying the criticism exactly to our manuscript. Still, we have shortened the paragraph a bit and weakened the emphasis on the discrepancies.

Lines 562-564: Remove "This can be seen …" to improve the readership.

**Response**: removed.

Lines 565-579: section 4.5.2. I do not think that this section is needed or adds any new information. The QBO influence on the stratospheric polar vortex is on the seasonal time-scale while the MJO influence is sub-seasonal. The mainly reason that the boreal winter signal of the MJO becomes stronger during eQBO is the MJO becomes more active, and deeper during eQBO. Thus, there is a stronger poleward propagation of wave activity during MJO phase 4, which induces more planetary waves entering the winter stratosphere, as it has been reported by Wang et al. (2018a).

**Response**: In other parts of the paper, reviewer 2 was missing physical evidence for statements, which we made based on statistical results. Here, we have discussed two physical possibilities (subsections 4.5.1 and 4.5.2, respectively), which we can statistically not distinguish (as stated in Sect. 4.5.3), but which could in principle both contribute to the statistically found QBO influence on the MJO-MA signal. We understand the criticism of reviewer 2 such that option 1 (subsection 4.5.1) should physically be the relevant one, so that the mentioning of option 2 can be omitted. We also think that option 1 is important, which is the reason why we have presented option 1 as actually the first option. However, we feel that excluding option 2 is not justified without further evidence for its irrelevance and must therefore be mentioned as possible mechanism as long as there is no particular reason against it. This becomes even more important as both mechanisms could actually contribute at the same time and the individual contribution strengths should be quantified in future. Only if such a quantification reveals that option 2 has indeed a neglectable contribution, this option could be safely dropped.

Line 615: "For the sake of brevity …" This statement should be in methodology or introduction section. Not in conclusion. If it has already stated, remove it.

**Response**: removed.

Line 635: no evidence provided for this, i.e. the MJO can lead to the initial planetary wave drag disturbances of the IHC mechanism. As far as I understand, part of the IHC mechanism involves

changes in gravity wave drag not just planetary waves. I would recommend remove this paragraph entirely as it has been discussed in Section 4 in detail.

**Response**: We have left the paragraph in the summary; however, we have removed any reference to an alteration of PW drag. Hence, we now only claim that we have presented statistical results that the MJO can trigger IHC, but without referring to the physical mechanism, which we have indeed not proven. With that the paragraph has also become shorter. In addition, we have called the last section "Summary and conclusion", so that a very condensed repetition of the discussion in Sect. 4 is intended. We just want to briefly mention that, of course, also gravity wave drag is important for the understanding of IHC. However, in the manuscript, we only refer to the initial disturbance, which starts the chain of dynamical effects und this initial disturbance is related to planetary waves.

Lines 655-659: This is not true. During winter, the stratospheric polar vortex descends with time climatologically. Once it is disturbed by a large-enough wave forcing, the associated warm temperature anomalies would also descend with time. On the timescale of the MJO, the descent of stratospheric anomalies in the polar region is thus fully expected as long as the initial wave drag is sufficiently large. The downward descent of polar vortex anomalies is one well-known pathway whereby the stratosphere influences the troposphere.

**Response**: We think that the general rejection of these lines by the reviewer 2 is based on a misunderstanding: Of course, we agree that a descending pattern is a well-known feature of the winter stratosphere. We wanted to put emphasis on the point that the descent has not been described in the context of IHC or the MJO influence and still think that some open questions are connected to this. In any case, we have rewritten the lines and weakened that statement to avoid such misunderstandings.

---

## Author Response (AR3)

**Response to the editor's comment.**

We would like to thank the editor Peter Haynes very much for handling the whole review process of our manuscript and appreciate its acceptance after the last iteration. We also thank Peter Haynes for suggesting language improvements in the final iteration, which we repeat in the following:

**Comment** by Peter Haynes on a possible correction before publication:

'Inner-stratospheric' -- you had previously used 'inner-tropospheric' and 'inner-stratospheric' but the former has now disappeared and 'inner-stratospheric' appears only twice. 'Inner-stratospheric' is an unusual term -- I don't recall it being used before in the literature -- and may cause confusion. I recommend replacing 'inner stratospheric' -- e.g.:

465: 'as well as the influence of the QBO, within the stratosphere, on high-latitude stratospheric dynamics'

477: 'The second possible type of interference is the influence, within the stratosphere, of the equatorial QBO':

**Response**: We very much appreciate your efforts to improve the manuscript also on a language level and have included both improvements as suggested.